# Skp2-mediated MLKL degradation confers cisplatin-resistant in non-small cell lung cancer cells

Huiling Zhou[1,2,6], Li Zhou[3,6], Qing Guan[1,2], Xuyang Hou[1,2], Cong Wang[2], Lijun Liu[1,2], Jian Wang[1,2], Xinfang Yu[4], Wei Li [5✉] & Haidan Liu [1,2✉]

Non-small cell lung cancer (NSCLC) is the most prevalent type of cancer and the leading cause of cancer-related death. Chemotherapeutic resistance is a major obstacle in treating NSCLC patients. Here, we discovered that the E3 ligase Skp2 is overexpressed, accompanied by the downregulation of necroptosis-related regulator MLKL in human NSCLC tissues and cell lines. Knockdown of Skp2 inhibited viability, anchorage-independent growth, and in vivo tumor development of NSCLC cells. We also found that the Skp2 protein is negatively correlated with MLKL in NSCLC tissues. Moreover, Skp2 is increased and accompanied by an upregulation of MLKL ubiquitination and degradation in cisplatin-resistant NSCLC cells. Accordingly, inhibition of Skp2 partially restores MLKL and sensitizes NSCLC cells to cisplatin in vitro and in vivo. Mechanistically, Skp2 interacts and promotes ubiquitination-mediated degradation of MLKL in cisplatin-resistant NSCLC cells. Our results provide evidence of an Skp2-dependent mechanism regulating MLKL degradation and cisplatin resistance, suggesting that targeting Skp2-ubiquitinated MLKL degradation may overcome NSCLC chemoresistance.

[1] Department of Cardiovascular Surgery, The Second Xiangya Hospital of Central South University, Changsha, Hunan, China. [2] Clinical Center for Gene Diagnosis and Therapy, The Second Xiangya Hospital of Central South University, Changsha, Hunan, China. [3] Department of Pathology, National Clinical Research Center for Geriatric Disorders, The Xiangya Hospital of Central South University, Changsha, Hunan, China. [4] Department of Medicine, Baylor College of Medicine, Houston, TX 77030, USA. [5] Department of Radiology, The Third Xiangya Hospital of Central South University, Changsha, Hunan, China. [6] These authors contributed equally: Huiling Zhou, Li Zhou. ✉email: Weililx@csu.edu.cn; haidanliu@csu.edu.cn

Non-small cell lung cancer (NSCLC), containing three major histological subtypes lung adenocarcinoma (LUAD), lung squamous cell carcinoma (LUSC), and large cell carcinoma, accounts for around 80-85% of lung cancer[1] and is a leading cause of cancer deaths worldwide[2]. Several actionable genetic alterations, such as *EGFR, ALK, ROS1, KRAS, c-MET, RET, NTRK, BRAF* and *HER2*, have been detected in patients with NSCLC and can be treated with agents for targeted therapies[3–5]. However, drug resistance to targeted therapy in this subgroup of patients inevitably occurs over time. After targeted therapies are exhausted, systemic therapy with chemotherapy, such as platinum-based chemotherapy, with or without incorporating immunotherapy using immune checkpoint inhibitors (ICIs), such as anti-programmed cell death protein ligand 1 (anti-PD-L1), anti-programmed cell death protein 1 (anti-PD-1), anti-cytotoxic T-lymphocyte-associated protein 4 (anti-CTLA-4) antibodies in the regimen is typically offered for care options. Further, resistance to platinum compounds (e.g., cisplatin and carboplatin) frequently occurs in NSCLC patients[6]. Thus, understanding the underlying mechanisms is necessary to overcome the obstacles against the clinical use of platinum compounds and finally cure cancers.

The E3 ligase, Skp2 (S-phase kinase-associated protein 2), is a crucial component of the SCF (Skp1-Cullin1-F-box) type of E3 ubiquitin-ligase complexes[7]. Skp2 functions as an oncoprotein and exerts oncogenic functions through ubiquitination of its substrates such as p21[8], p27[9], p57[10], E-cadherin[11], FOXO1[12], Akt[13], ING3[14], and others. Therefore, Skp2 governs many critical cellular processes, including cell growth, apoptosis, differentiation, cell cycle progression, migration, invasion, and metastasis[15]. High Skp2 expression has been found in various human cancer[15,16], including NSCLC[17,18], and is associated with lymph node metastasis, vascular invasion, and poor patient survival. Additionally, Skp2 has reportedly developed drug resistance in various human cancers[19]. It has been demonstrated that defective regulation of Skp2 is linked to rapamycin resistance in different human tumor cell lines[20]. Skp2 is associated with resistance to preoperative doxorubicin-based chemotherapy[21], paclitaxel[19], and gefitinib[22] resistance. High level of Skp2 confers cisplatin resistance in nasopharyngeal carcinoma cells[23] and mantle cell lymphoma cells[24]. Upregulation of Skp2 reportedly enhanced cisplatin resistance in A549 cells[25]. However, the role of Skp2 in cisplatin resistance of NSCLC has not been fully elucidated.

The *MLKL* (mixed-lineage kinase domain-like) gene plays a crucial role in necroptosis execution[26–28]. Briefly, RIPK3-mediated phosphorylation of MLKL triggers a conformational change that facilitates the translocation to, and eventual irreversible disruption of, cellular membranes, finally leading to necroptotic cell death[29]. Low expression of MLKL protein is associated with decreased overall survival (OS) in patients with pancreatic adenocarcinoma[30], colon cancer[31], ovarian cancer[32] and NSCLC[33], probably due to insufficient MLKL necroptosis signaling, suggesting that necroptosis is an important determinant of cancer cell death and outcomes in these patients. However, the relevant reports about the relationship between MLKL and chemoresistance are unclear.

Cisplatin is one of the most effective chemotherapeutic drugs to treat human cancers, including NSCLC[34]. Unfortunately, many patients develop drug resistance during cisplatin-based treatment. In this study, we established cisplatin-resistant NSCLC cell lines and investigated whether Skp2 has a pivotal role in NSCLC cisplatin resistance. We identified Skp2 as an E3 ligase that ubiquitinates MLKL. Skp2-regulated MLKL ubiquitination and degradation, at least partially, contributes to cisplatin-resistance in NSCLC cells. Genetic or pharmacological inactivation of Skp2 re-sensitizes the cisplatin-resistant NSCLC cells toward cisplatin treatment, suggesting that MLKL ubiquitination by Skp2 is a potential therapeutic target to overcome NSCLC chemoresistance.

## Results

**Skp2 is highly expressed and associated with tumorigenic properties of human non-small cell lung cancer.** We first examined the protein level of Skp2 by Western blotting in immortalized non-tumor cell HBE and MRC5 and a panel of human NSCLC cell lines. The result showed that except the H460 and H358 cells, Skp2 was highly expressed in almost all tested human NSCLC cell lines (Fig. 1a, b). We further determined Skp2 levels in the NSCLC tissues and paired adjacent non-tumor tissues. The data demonstrated that the expression of Skp2 is significantly increased in NSCLC tissues compared with the adjacent tissues (Fig. 1c, d, Supplementary Table 1). To assess the effect of Skp2 on the proliferation of NSCLC cells, we generated Skp2 stable knockdown H1299, H23, H125, and A549 cell lines and validated shRNAs that effectively blunted *Skp2* expression after transfection (Fig. 1e). Compared with the shGFP control, knockdown of *Skp2* inhibited cell proliferation (Fig. 1f) and anchorage-independent cell growth (Fig. 1g) of these cell lines. In order to determine the role of Skp2 in the tumorigenesis of NSCLC in vivo, we conducted athymic nude mouse models. We found that inhibition of Skp2 expression significantly suppressed tumor growth in the H23 xenograft model (Fig. 1h–j). These results indicate that Skp2 plays a critical role in NSCLC tumorigenesis, and knockdown of *Skp2* in NSCLC cells reduces tumorigenic properties.

**Skp2 protein is inversely correlated with MLKL protein in NSCLC.** We found that NSCLC cells with higher Skp2 expressed lower MLKL, whereas NSCLC cells with lower Skp2 had higher MLKL (Fig. 1a, b). Skp2 and MLKL showed opposite expression levels in NSCLC cell lines. Consistently, as detected by immunohistochemistry, in most cases with reduced MLKL protein, a striking association with increased Skp2 levels was observed and the expression of MLKL is significantly decreased in the NSCLC tissues as compared with the adjacent tissues (Fig. 1c, d, Supplementary Table 2). Skp2 expression level was inversely correlated with MLKL in NSCLC clinical samples (Fig. 2a–d). To further extend our observations to a clinicopathologically relevant context, we performed a Kaplan–Meier survival analysis of Skp2 and MLKL in 813 NSCLC patients. The results showed high *Skp2* mRNA expression was associated with worse overall survival (OS). On the contrary, high *MLKL* mRNA expression was significantly associated with a favorable prognosis (Fig. 2e). These findings indicate that Skp2 negatively regulates MLKL protein levels and suggest that MLKL is a novel substrate of Skp2. The Skp2-MLKL axis plays an essential role in NSCLC development and correlates with the prognosis of patients with NSCLC.

**Altered Skp2 expression or activity impacts the protein level of MLKL.** To determine whether Skp2 regulated the protein level of MLKL directly, 293T cells were transiently co-transfected with Skp2 and MLKL. The results indicated that co-expression of Skp2 decreased MLKL dose-dependently (Fig. 3a). Furthermore, ectopically increasing Skp2 expression also caused a reduction of endogenous MLKL levels (Fig. 3b) in both H358 and H460 cells with relatively low detectable Skp2 protein but a high level of MLKL (Fig. 1a). These results indicated that Skp2 reduces MLKL protein exogenously and endogenously in a dose-dependent manner. Knockdown of *Skp2* by shRNA increased the endogenous MLKL protein, accompanied by an elevation of p27 protein, a well-known Skp2 substrate, in H1299, A549, and H23 cells (Fig. 3c). Skp2 inhibitor SZL P1-41, which selectively suppresses

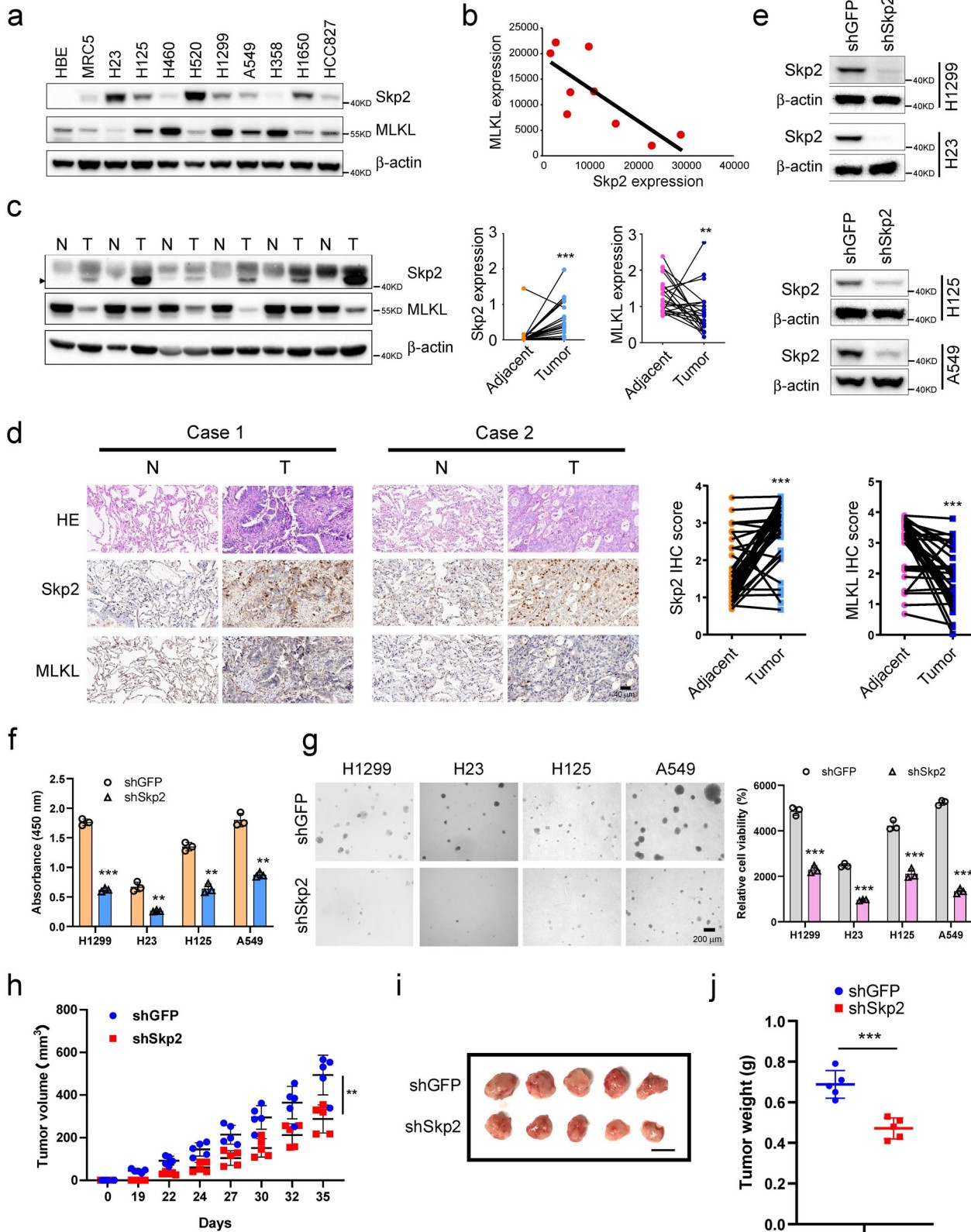

Skp2 E3 ligase activity but not the activity of other SCF complexes[35], resulted in the accumulation of MLKL and p27 in A549 and H1299 cells (Fig. 3d–f). Notably, the MLKL mRNA level upon Skp2 knockdown was comparable to that in control cells (Fig. 3g). These results indicate that Skp2 regulates MLKL abundance, at least partly, through post-translational modifications.

**Skp2 interacts with MLKL.** Next, we assessed whether Skp2 interacts with MLKL by co-immunoprecipitation assays. GFP-tagged MLKL and Flag-tagged Skp2 expression vectors were transfected either alone or together into 293T cells and co-immunoprecipitation was performed using an anti-Flag or anti-GFP antibody. The results showed that Flag-Skp2 was detected in the GFP-MLKL immunoprecipitants (Fig. 4a). GFP-MLKL was

**Fig. 1 Expression of Skp2 and MLKL in human non-small cell lung cancer. a** Western blot analysis examined Skp2 and MLKL expression in several NSCLC cell lines, human normal lung fibroblast cell line MRC5 and immortalized human bronchial epithelial cell line HBE. β-actin was used as a loading control. **b** Scatterplot shows a negative correlation between the expression of Skp2 and MLKL in human NSCLC cell lines. Pearson's coefficient tests were performed to assess statistical significance. **c** Skp2 and MLKL protein levels in six representative NSCLC cases was assessed by Western blot analysis. β-actin was used as a loading control (*left panel*). N, adjacent non-tumor tissue; T, tumor. Western blotting determined Skp2 (*middle panel*) and MLKL (*right panel*) protein levels in the malignant and the corresponding normal adjacent tissues of 22 NSCLC patients. The intensity was evaluated using Image J (NIH) computer software. $N = 22$, **$p < 0.01$, ***$p < 0.001$ by Student's $t$ test, significant difference between groups as indicated. **d** Representative images of immunohistochemical staining for Skp2 and MLKL in represent specimens among 39 cases of human NSCLC (*left panel*). IHC scores for Skp2 (*middle panel*) and MLKL (*right panel*) in 39 paired tumors and adjacent tissues. $N = 39$, ***$p < 0.001$ by Mann–Whitney $U$ test. **e–g** Knockdown of Skp2 (**e**) attenuated NCI-H1299, NCI-H23, NCI-H125 and A549 cell proliferation ($n = 3$) (**f**) and anchorage-independent cell growth in soft agar ($n = 3$) (**g**). Data represent mean ± SD from three independent experiments. **$p < 0.01$, ***$p < 0.001$ by Student's $t$ test, significant difference compared with the shGFP control cells. **h–j** Knockdown of Skp2 suppresses tumor growth in vivo. Average tumor volume (**h**), photographed xenograft tumors (**i**) and average tumor weight (**j**) of H23-shGFP and H23-shSkp2 xenografts were shown. Data are shown as mean values ± S.D. $N = 5$ mice per group, ***$p < 0.001$ by Student's $t$ test, significant difference compared with the shGFP control group.

also found in the Flag-Skp2 immune complex (Fig. 4b). We then sought to determine whether endogenous MLKL interacts with Skp2 in H1299 cells. MLKL and Skp2 were separately immuno-precipitated from H1299 cells, and the reciprocal protein was detected by Western blotting. As shown in Fig. 4c, d, both Skp2 and MLKL were detected in their individual immunoprecipitated complexes but not in the isotype-matched negative control IgG complexes. Endogenous MLKL and Skp2 interaction was also found in A549 cells (Fig. 4e, f). Moreover, the Skp2 inhibitor SZL P1-41 blocked the Skp2-MLKL association in A549 cells (Fig. 4g). Thus, the results indicate that MLKL and Skp2 have physical contact at both exogenous and endogenous levels.

**Skp2 regulates MLKL protein stability and degradation through ubiquitination.** Following the ectopic expression of Skp2 decreased MLKL protein levels (Fig. 3a, b), we assessed whether Skp2 regulates MLKL stability. The endogenous MLKL protein was stabilized by treatment with the proteasome inhibitor MG132 in H1299 and H23 cells (Fig. 5a, Supplementary Fig. 1a, lane 1 vs lane 3). The MG132 treatment masked the effect of Skp2 depletion in both H1299 and H23 cells (Fig. 5a, Supplementary Fig. 1a, lane 2 vs lane 4). Ectopic Skp2 expression decreased endogenous MLKL protein level, which was restored by treatment with the proteasome inhibitor MG132 in A549 cells (Fig. 5b). The results suggest that MLKL is degraded by the 26S proteasome. MLKL half-life was determined using a cycloheximide (CHX) chase experiment. The results demonstrated that knockdown of Skp2 extended MLKL protein half-life (Fig. 5c, Supplementary Fig. 1b), suggesting that Skp2 destabilized MLKL protein. Since Skp2 is an E3 ligase for targeted proteins to contribute to proteasomal-mediated degradation through ubiquitination, we thus determined whether MLKL is a substrate for Skp2-induced ubiquitination. To test this hypothesis, we co-transfected 293T cells with *His-Ub*, *GFP-MLKL* and *Flag-Skp2* plasmids, and ubi-quitinated MLKL was subjected to SDS-PAGE analysis using an anti-GFP antibody. The results show that MLKL ubiquitination increased by Skp2 overexpression compared to the vector control (Fig. 5d), whereas MLKL ubiquitination decreased by SZL P1-41-mediated Skp2 inhibition (Fig. 5d). Knockdown of *Skp2* sig-nificantly attenuated endogenous MLKL ubiquitination in both H1299 (Fig. 5e) and H23 (Supplementary Fig. 1c) cells. These lines of evidence suggest that Skp2 regulates MLKL through ubiquitin-mediated proteasomal degradation.

**Skp2-ubiquitinated MLKL degradation confers cisplatin-resistant in NSCLC cells.** Given the regulation of MLKL stabi-lity by Skp2, we asked whether Skp2 and MLKL expression is altered in cisplatin-resistant cells. H1299 and A549 cells were

induced by continuous exposure and gradually increasing con-centrations of cisplatin for more than 6 months to establish resistant cells. The cell viability assay showed that 20 μM cisplatin led to more than 50% cell growth inhibition in both H1299 and A549 cells. However, the cisplatin-resistant H1299R and A549R cells resisted the growth inhibitory properties of 20 μM cisplatin (Fig. 6a). Skp2 was obviously upregulated in resistant cells com-pared to the parental cell lines. In contrast, MLKL, RIP1 and RIP3 were downregulated (Fig. 6b). To determine whether a high level of Skp2 is required to maintain cisplatin resistance, Skp2 siRNAs were transfected in cisplatin-resistant H1299R and A549R cells to knockdown Skp2 expression. The results indicated that siRNA-mediated *Skp2* depletion was accompanied by an increased MLKL protein in both H1299R and A549R cells (Fig. 6c). Cisplatin substantially attenuated cell viability in the parental H1299 and A549 cells irrespective of the Skp2 status. However, the significant inhibitory effect was only observed in *Skp2*-knockdown but not in siCtrl-H1299R (Fig. 6d) and siCtrl-A549R cells (Fig. 6e). Upon acutely exposure to cisplatin, the Skp2 protein level decreased was accompanied by the MLKL protein level slightly increased in parental A549 cells, whereas Skp2 and MLKL protein levels did not obviously change in A549R cells (Supplementary Fig. 2). We also assessed endogenous MLKL and Skp2 interaction in the presence/absence of cisplatin via Co-IP in both A549R and A549 cells. The endogenous association between MLKL and Skp2 was diminished in both A549R and A549 cells treated with cisplatin (Fig. 6f, g). Moreover, ectopic MLKL expression enhanced cisplatin-reduced cell viability in both H1299R and A549R cells (Fig. 6h, i). To determine whether the chemoresistance was associated with Skp2-mediated MLKL downregulation in NSCLC cells, we examined the efficacy of the Skp2 inhibitor, SZL P1-41, in abrogating cisplatin resistance. Although cisplatin and SZL P1-41 alone partially decreased cell viability in H1299R and A549R cells, the combination therapy enhanced the anti-tumor effects (Fig. 6j, k). The results indicated that genetic or pharmacologic recovery of MLKL attenuates cisplatin resistance in NSCLC cells. Collectively, these results suggest that Skp2 overexpression mediates MLKL ubiquitination and degradation involved in cis-platin resistance in NSCLC.

**Skp2-ubiquitinated MLKL degradation confers cisplatin-resistant of NSCLC cells in vivo.** Since the upregulation of Skp2 was accompanied by the downregulation of necroptosis-related regulators MLKL, RIP1 and RIP3 in cisplatin-resistant H1299R and A549R cells (Fig. 6b), we further investigated the ubiquitination status of MLKL in cisplatin-resistant cells. The results demonstrated that compared with the parental A549 cells, a significant increase of MLKL ubiquitination in cisplatin-resistant A549R cells (Fig. 7a). Notably, pre-treatment with

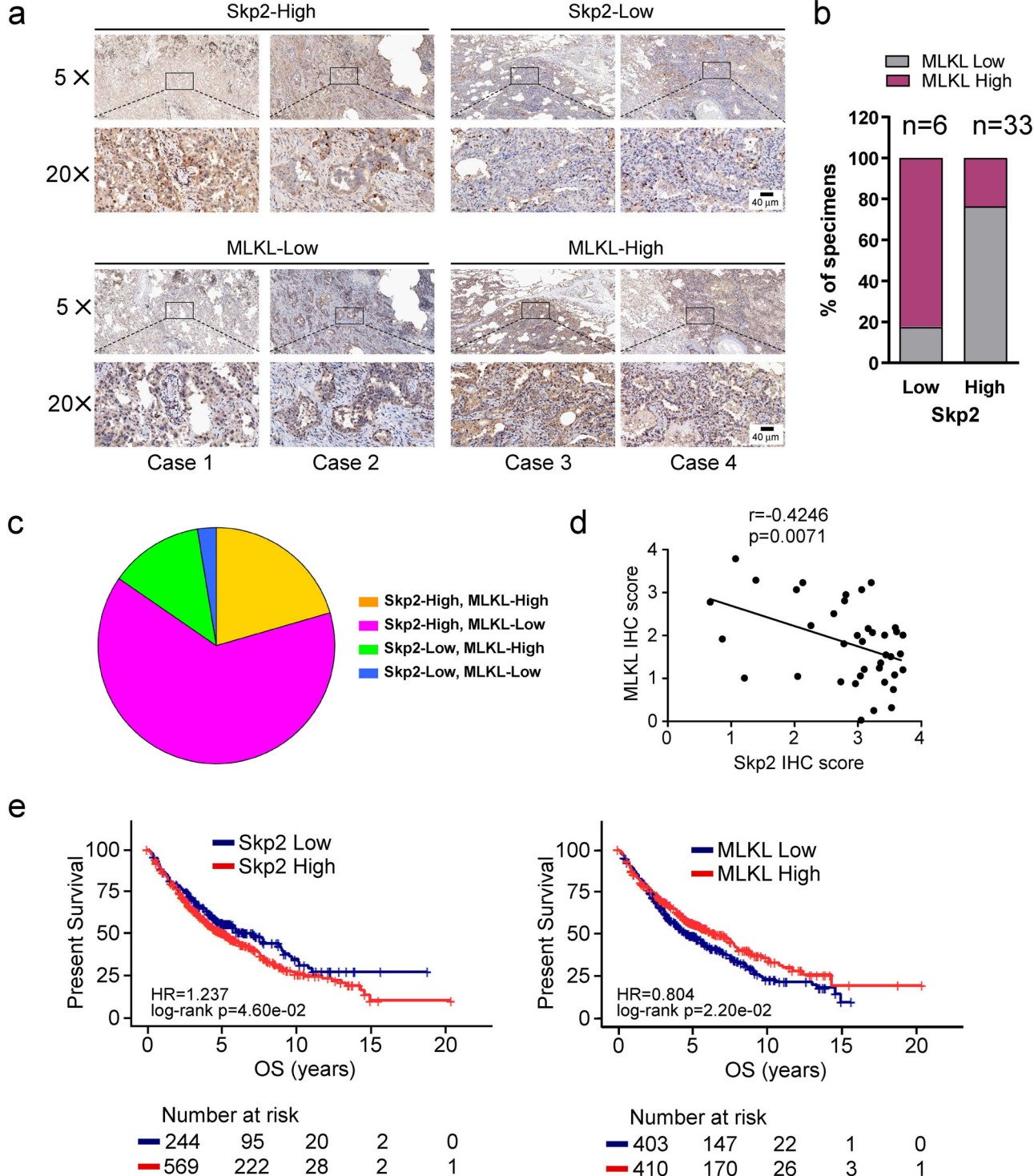

**Fig. 2 Skp2 expression is inversely correlated with MLKL in NSCLC tissues. a** Representative cases from 39 NSCLC specimens were analyzed by IHC staining with Skp2 and MLKL. **b** The percentage of samples displaying low or high Skp2 expression compared to the expression levels of MLKL. $N = 39$. **c** The relative proportions of protein expressions were illustrated as a pie chart. **d** Scatterplot showed a negative correlation between Skp2 and MLKL. **e** Kaplan–Meier analysis showing correlation between Skp2 or MLKL expression levels and overall survival (OS) of patients ($n = 813$) with NSCLC. The gene expression data and survival information were downloaded from the Gene Expression Omnibus (GEO) database, including GSE29013, GSE37745, GSE31210, and GSE157011 datasets, based on GPL570 platform. The plots were draw by the R software. $p < 0.05$ by log-rank (Mantel–Cox) test was considered to be a statistically significant difference.

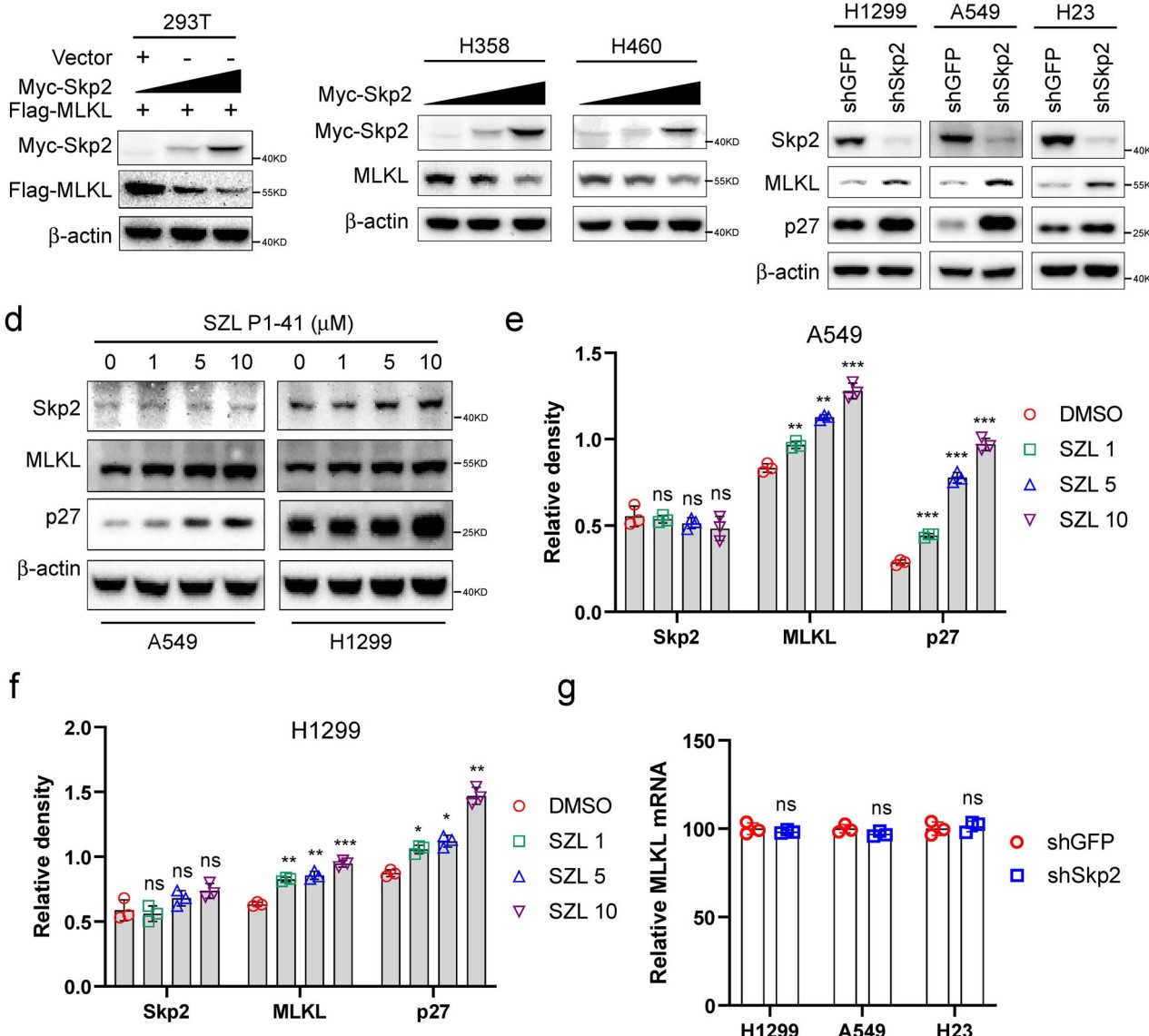

**Fig. 3 Skp2 negatively regulates MLKL protein level. a** Transient expression of Skp2 reduced Flag-MLKL protein. 293T cells were cotransfected with the constructs as indicated. Whole cell extracts (WCEs) were subjected to Western blot analysis. **b** Overexpression of Skp2 reduces endogenous MLKL protein level. H358 and H460 NSCLC cells were cotransfected with different doses of Skp2 expression construct, WCEs were subjected to Western blot analysis for endogenous MLKL protein level. **c** Knockdown of Skp2 resulted in an increased level of endogenous MLKL protein. Stable knockdown of endogenous Skp2 by shRNA in H1299, A549 and H23 NSCLC cells, WCEs were prepared for Western blot analysis. **d–f** Pharmacologic inhibition of Skp2 accumulated the MLKL protein level. A549 and H1299 cells were treated with indicated doses of Skp2 inhibitor SZL P1-41 for 24 h, WCEs were subjected to Western blot analysis. The intensity was evaluated using Image J (NIH) computer software (**e**, **f**). Data represent mean ± SD from three independent experiments. ns, not statistically significant. *$p < 0.05$, **$p < 0.01$ by 1-way ANOVA test with Dunnett's multiple comparisons test, significant difference compared with the DMSO-treated group. **g** MLKL mRNA expression in Skp2 silencing NSCLC cells was examined by RT-qPCR. Data represent mean ± SD from three independent experiments. ns by Student's t test, not statistically significant.

necroptosis inhibitor Necrostatin-1 (Nec-1) significantly compromised cisplatin-reduced cell viability in *Skp2*-knockdown (Fig. 7b) and SZL P1-41-treated (Fig. 7c) A549R cells, suggesting that necroptosis could be involved in NSCLC cells' resistance to cisplatin. The WB result showed that Nec-1 significantly abolished the phosphorylation of RIPK1 at Ser166 in A549 cells, indicating that Nec-1 is functioning appropriately (Supplementary Fig. 3). Further, transient knockdown of *MLKL* by siRNA compromised cisplatin-reduced cell viability in both H1299R-sh*Skp2* and A549R-sh*Skp2* stable cells, confirming the critical role of MLKL in cisplatin resistance in NSCLC (Fig. 7d, e). Xenograft tumors derived from *Skp2*-knockdown A549R cells treated with

cisplatin exhibited a significant decrease in growth and weight compared to tumors derived from shGFP-A549R cells treated with cisplatin (Fig. 7f–h). The IHC data revealed that xenograft tumors from cisplatin-treated *Skp2*-knockdown A549R cells exhibited a significant decrease in the protein level of Skp2, Ki67 and elevation of MLKL protein and phosphorylation at Thr357, which is critical for necroptosis[36], when compared to cisplatin-treated shGFP-A549R xenograft tumors (Fig. 7i, j). Together, these results suggest that the Skp2-ubiquitinated MLKL degradation plays a critical role in cisplatin resistance and that combining cisplatin with Skp2 inhibitors could be a promising strategy to overcome chemoresistance in NSCLC cells.

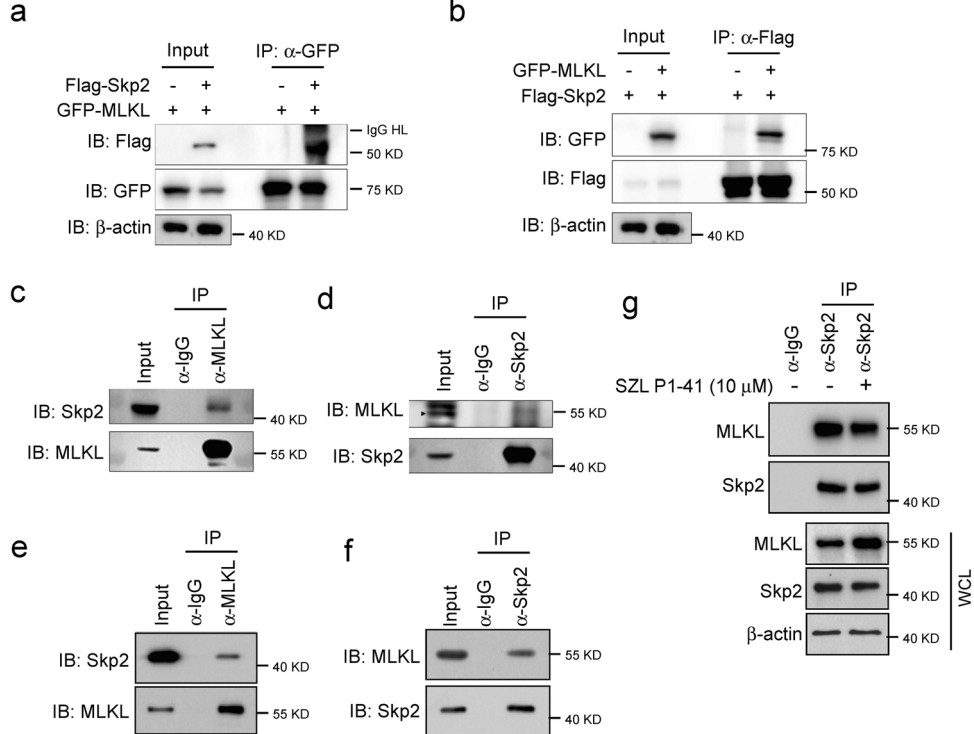

**Fig. 4 Skp2 interacts with MLKL. a, b** Co-immunoprecipitation (Co-IP) analysis of Skp2 and MLKL interaction. 293T cells were transfected with the constructs as indicated, WCEs were subjected to Co-IP assay and Western blot analysis. **c, d** Co-IP analysis of endogenous Skp2 and MLKL interaction in H1299 cells. WCEs were collected and subjected to Co-IP assay and Western blot analysis. **e, f** Co-IP analysis of endogenous Skp2 and MLKL interaction in A549 cells. WCEs were collected and subjected to Co-IP assay and Western blot analysis. **g** A549 cells were treated with the indicated dose of Skp2 inhibitor SZL P1-41 for 24 h, WCEs were subjected to Co-IP assay and Western blot analysis.

## Discussion

It is well established that Skp2 requires prior phosphorylation of its targets for destruction. Most Skp2 targets are recognized when phosphorylated[7]. For example, p27 phosphorylation on Thr187 by CDKs[37], FOXO1 phosphorylation on Ser256 by Akt[12], or RASSF1A phosphorylation on Ser203 by cyclinD-CDK4[38] is required for Skp2-mediated ubiquitination and degradation. Skp2 interacts and promotes the ubiquitination and destruction of Casein Kinase I (CKI)-phosphorylated E-cadherin. Mutating the CKI-phosphorylated sites abolishes Skp2-mediated E-cadherin ubiquitination and degradation and significantly reduces cellular migration ability[11]. Phosphorylation of PDCD4 on Ser67 by Akt is a requirement for Skp2-regulated PDCD4 ubiquitination[39]. Multiple phosphorylation sites have been identified on MLKL, including RIPK3-mediated phosphorylation on Ser345, Ser347 and Thr349 in mice, and Thr357 and Ser358 in humans[26,36,40]. Another site in human MLKL, corresponding to mouse MLKL Ser124, was phosphorylated by an unknown kinase in a cell-cycle-dependent manner during the G1- to M-phase transition[41,42]. Additionally, Ser228 and Ser248 in mouse MLKL are subject to RIPK3-mediated phosphorylation[43]. Recently, members of the TAM (Tyro3, Axl, and Mer) family of receptor tyrosine kinases have been identified to phosphorylate Tyr376 of MLKL under necroptotic stress, which promotes MLKL oligomerization[44]. Our result showed that Skp2 promoted MLKL ubiquitination (Fig. 5d) and Skp2 inhibitor SZL P1-41 blocked MLKL ubiquitination (Fig. 5d), suggesting that Skp2 participated in the regulation of MLKL polyubiquitination. However, whether MLKL needs prerequisite phosphorylation by upstream kinase and the mechanisms underlying Skp2 recognition of MLKL need further investigation. Moreover, high Skp2 mRNA or low MLKL mRNA expression in NSCLC tumor tissues was associated with worse

overall survival (OS) by an online tool analysis (Fig. 2e). Although the integration of various data sets increased sample size, the prognostic value originates from different patient cohorts with largely different survival/gene expression may cause the results bias. In addition, more evidence is required to clarify the regulation mechanism between them and their clinical values in protein levels because our results indicated that Skp2 regulated MLKL protein in a ubiquitination-dependent manner.

Skp2 can conjugate two conventional K48- and K63-linked polyubiquitination chains to different targets. The K48-linked ubiquitination by Skp2 regulates protein stability and leads to proteasome-mediated proteolysis of targeted proteins. For instance, Skp2 induces K48-linked polyubiquitination of p27[45], FOXO1[12], mH2A1[46], and PDCD4[39] followed by proteasome-mediated degradation. In addition, the nonproteolytic Skp2 mediates K63-linked polyubiquitination of substrates to exert other regulation functions, including trafficking, subcellular localization, and kinase activation. Akt is an Skp2 substrate undergoing K63-linked polyubiquitination, which leads to Akt activation and membrane recruitment[13]. Tenascin-C is K63-ubiquitinated by Skp2, particularly at K942 and K1882, thus promoting its recognition by p62 and its selective autophagic degradation[47]. Skp2 directly interacts with Aurora B and triggers Aurora B K63-linked ubiquitination to regulate Aurora B activation in cell mitosis and spindle checkpoint[48]. Skp2 interacts with NBS1 and triggers K63-linked ubiquitination of NBS1 during DNA double-strand breaks (DSBs), promoting the interaction of NBS1 with ATM, which is critical for ATM activation in response to DNA damage[49]. Apart from the substrates mentioned above, Skp2 also induces K63-linked ubiquitination of many other substrates, such as Akt[13], Twist[50], LKB1[51], YAP[52], RagA[53], Bcr-Abl[39] and MTH1[54]. Our result showed that Skp2 promoted

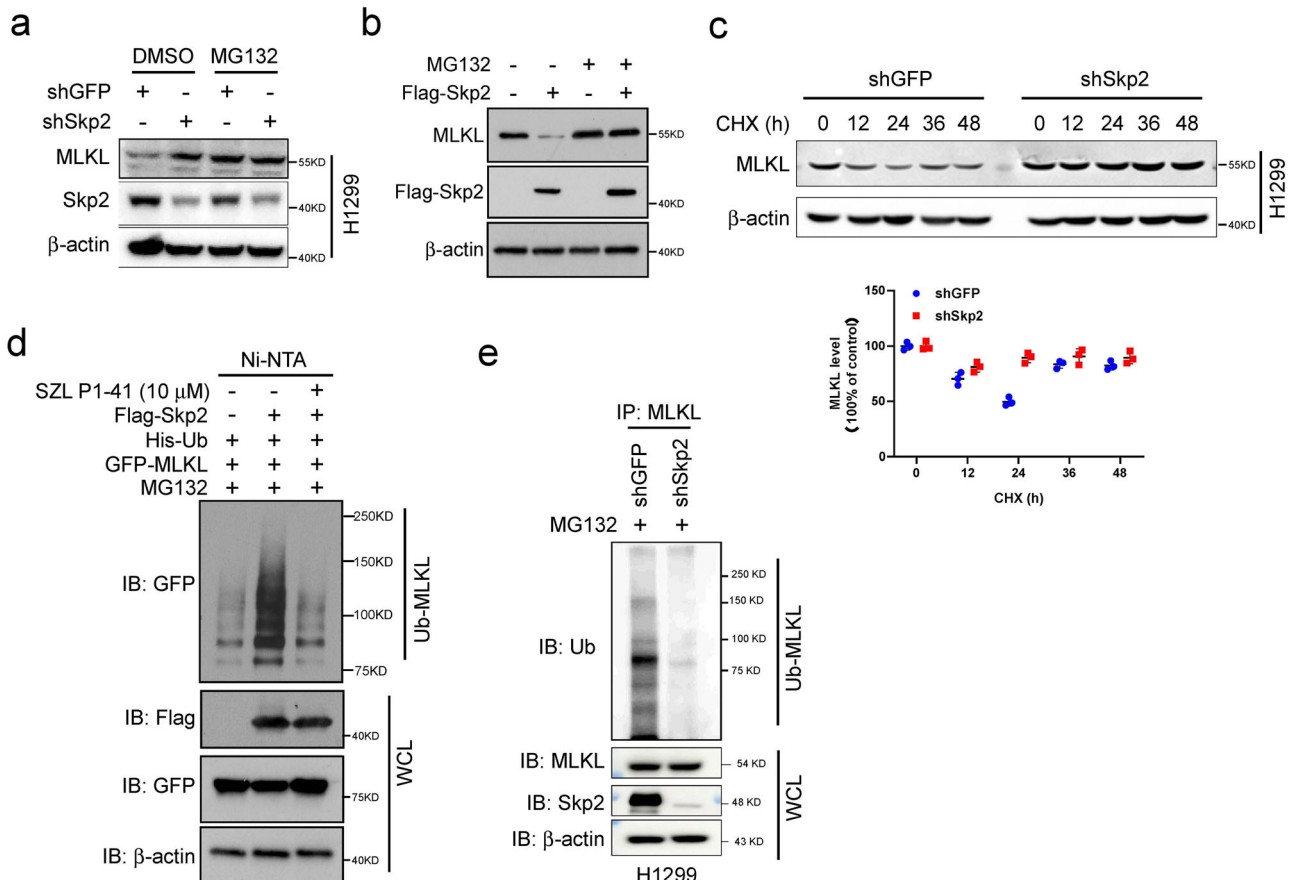

**Fig. 5 Skp2 regulates the stability and ubiquitination of MLKL protein. a** MLKL expression was analyzed in H1299 cells expressing shGFP or shSkp2 and treated with MG132 (20 μM) for 12 h. WCEs were subjected to Western blot analysis. **b** Ectopically expressed Skp2 and treated with MG132 (20 μM) for 12 h in A549 cells. WCEs were subjected to Western blot analysis. **c** H1299 cells expressing shGFP or shSkp2 were subjected to cycloheximide chase (25 μg/mL for indicated times). WCEs were subjected to Western blot analysis (top) and the protein level was qualified (bottom). Data represent mean ± SD from three independent experiments. **d** 293T cells transfected with the indicated plasmids and cultured in the presence of DMSO or compound SZL P1-41 (10 μM) for 24 h, then treated with MG132 (20 μM) for 6 h followed by in vivo MLKL ubiquitination assay. **e** Stable Skp2 knockdown H1299 cells were treated with MG132 (20 μM) for 6 h, WCEs were prepared and subjected to MLKL ubiquitination analysis using an ubiquitin antibody.

MLKL ubiquitination (Fig. 5d, e, Supplementary Fig. 1c) and the endogenous MLKL protein was stabilized by the proteasome inhibitor MG132 treatment (Fig. 5a, Supplementary Fig. 1a, lane 1 vs lane 3). Ectopic Skp2 expression decreased endogenous MLKL protein level, which was restored by treatment with the proteasome inhibitor MG132 (Fig. 5b). These results suggested that ubiquitinated-MLKL undergoes proteasome-dependent degradation. K48-linked ubiquitination is the most abundant Ub chain type targeting proteins for proteasomal degradation. Nevertheless, a previous report showed that all non-K63-linked ubiquitination could result in the proteasomal degradation of proteins[55]. Recent reports show that K6-linked[56], and K11-linked[57] ubiquitination could result in proteasomal degradation of proteins.

Knockdown of Skp2 is more effective in increasing cisplatin cytotoxicity in cisplatin-resistant cells, suggesting Skp2 is likely to be a more promising target in treating cisplatin-resistant tumor cells[24]. Similarly, Skp2-deficiency augmented CRPC cells' sensitivity to doxorubicin or paclitaxel treatment. Moreover, doxorubicin or paclitaxel, in conjunction with Skp2 inhibitor compound#25 (SZL P1-41) substantially heightened the cytotoxicity of CRPC cells[50]. Recent studies indicate that Skp2 deficiency renders Her2-positive breast cancer cells more sensitive to Herceptin treatment and prolongs the survival of Her2-positive patients, highlighting that Skp2 is an appealing therapeutic target

to combine with Herceptin for cancer treatment[13]. In addition, genetic ablation or pharmacological inactivation of Skp2 re-sensitizes the gefitinib-resistant NSCLC cells toward gefitinib treatment, improving the potential of Skp2 targeting to overcome gefitinib resistance in NSCLS patients[22]. In line with these reports, our results demonstrated that a significant inhibitory effect of cisplatin was only observed in Skp2-knockdown but not in control-knockdown H1299R (Fig. 6d) and A549R cells (Figs. 6e, 7b, 7f–h). Our results indicated that knockdown of Skp2 induces p27 expression (Fig. 3c). The M phase and G2 phase are the most sensitive stages to radiotherapy or DNA damage agents. Knockdown of Skp2 induces p27 expression and cell cycle arrest which may enhance the anti-tumor effect of cisplatin by DNA damage. Efforts have been made to develop novel inhibitors based on different strategies, and combining these agents with conventional chemotherapy appears to be an attractive approach for treating chemotherapy-resistant cancers.

The oncogenic property of Skp2 made it an appealing pharmacological target for cancer prevention and treatment. In recent years, several specific inhibitors for Skp2 have been developed. SZL-P1 41 is identified to physically bind to Skp2, thus preventing Skp2 SCF complex formation and inhibiting Skp2 E3 ligase activity, suppressing cancer cells' survival both in vitro and in vivo[35]. Compound A (CdpA) inhibits Skp2-mediated p27 ubiquitination and Skp2 E3 ligase activity by excluding Skp2 from

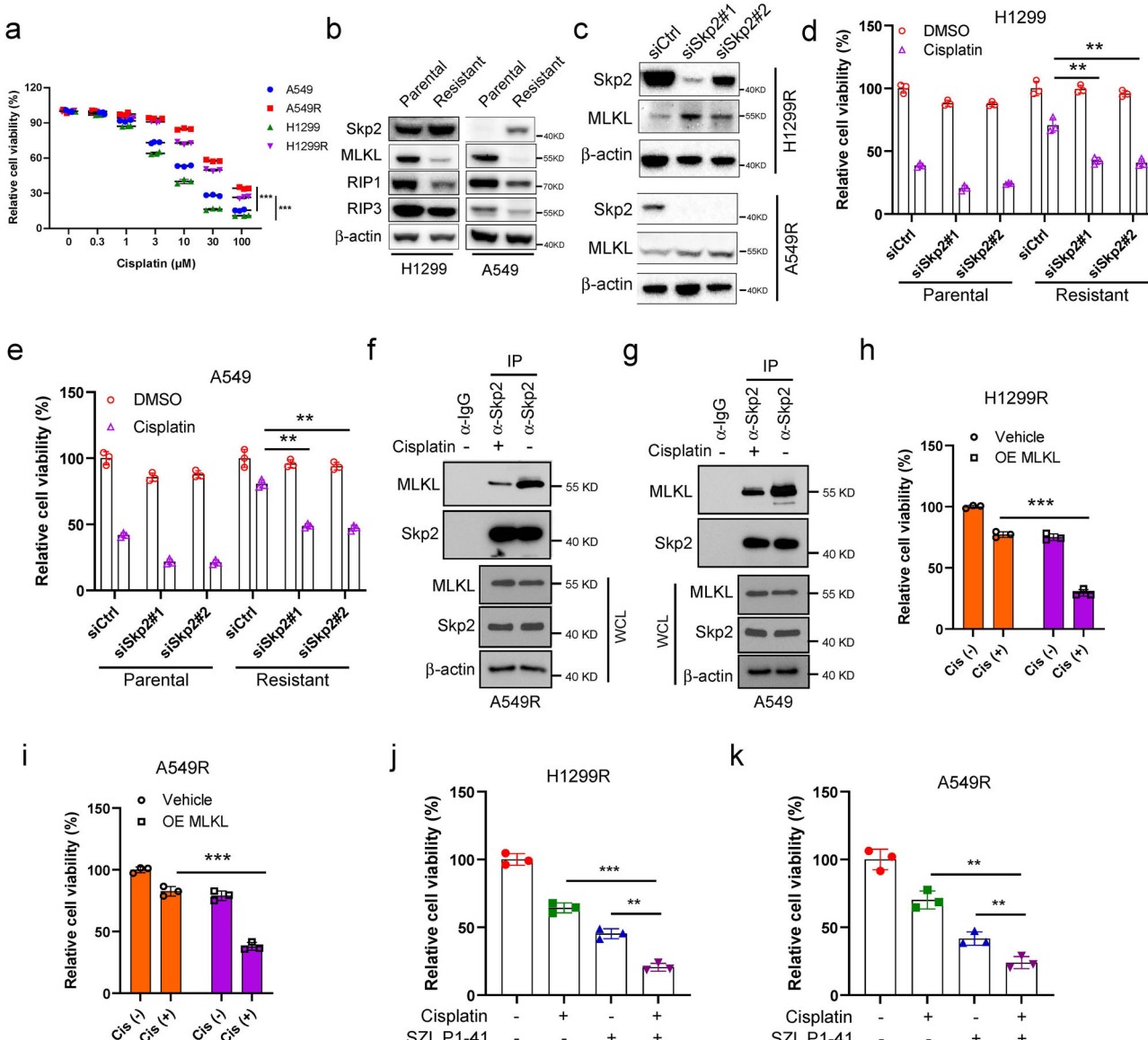

**Fig. 6 Skp2-ubiquitinated MLKL degradation confers cisplatin-resistant in NSCLC cells. a** Comparison of proliferation inhibition rate of H1299/H1299R and A549/A549R cells at the indicated concentrations of cisplatin for 72 h. Data represent mean ± SD from three independent experiments. $p < 0.001$ by Student's $t$ test. **b** Western blot analysis showed the protein level of Skp2, MLKL, RIP1 and RIP3 in established cisplatin-resistant (H1299R and A549R) cells and parental (H1299 and A549) cells. **c–e** Silencing of Skp2 enhances the sensitivity of cisplatin-resistant NSCLC cells to cisplatin. The control or Skp2 siRNA was transiently transfected into H1299R and A549R cells for 72 h. Knockdown of Skp2 was confirmed by Western blot analysis (**c**). The parental (H1299 and A549) and cisplatin-resistant (H1299R and A549R) cells were transiently transfected for 24 h with the control or Skp2 siRNA, trypsinized, and transferred to a 96-well plate without or with cisplatin (20 μM) for 48 h, followed by cell viability assay (**d**, **e**). Data represent mean ± SD from three independent experiments. ***, $p < 0.001$ by 1-way ANOVA test with Dunnett's multiple comparisons test, significant difference between groups as indicated. **f**, **g** The cisplatin-resistant A549R (**f**) and parental A549 (**g**) cells were treated without/with cisplatin (20 μM) for 48 h, WCEs were collected and subjected to Co-IP assay and Western blot analysis. **h**, **i** Ectopically expressed MLKL following cisplatin treatment and conducted cell viability assays in H1299R and A549R cells. Data represent mean ± SD from three independent experiments. ***$p < 0.001$ by Student's $t$ test, significant difference compared with the cisplatin-treated Vector-tansfected group. **j**, **k** H1299R and A549R cells treated with the vehicle control, cisplatin (20 μM), the Skp2 inhibitor SZL P1-41 (10 μM), or cisplatin and SZL P1-41 combination for 48 h, followed by cell viability assays. Data represent mean ± SD from three independent experiments. **$p < 0.01$, ***$p < 0.001$ by 1-way ANOVA test with Dunnett's multiple comparisons test, significant difference compared with the cisplatin-treated group or the SZL P1-41-treated group.

the SCF complex, thus inducing G1/S cell-cycle arrest and tumor cell killing[58]. Compounds NSC689857 and NSC681152 disrupt the interaction between Skp2 and Cks1, which inhibits Skp2 from recognizing Thr187 phosphorylated p27 and subsequent ubiquitination[59]. Compound SMIP0004 has been found to restrain Skp2 expression, thus protecting p27 from degradation to induce p27 accumulation, finally inhibiting prostate cancer cell

growth and inducing apoptosis[60]. Although the efficacy of these compounds for human cancer needs to be further justified, Skp2 is still an attractive target for cancer therapy. Developing an Skp2-specific inhibitor with low side effects would benefit cancer therapy.

Impairment of cell death pathways or resistance to cell death is a hallmark of various cancers, which is the main

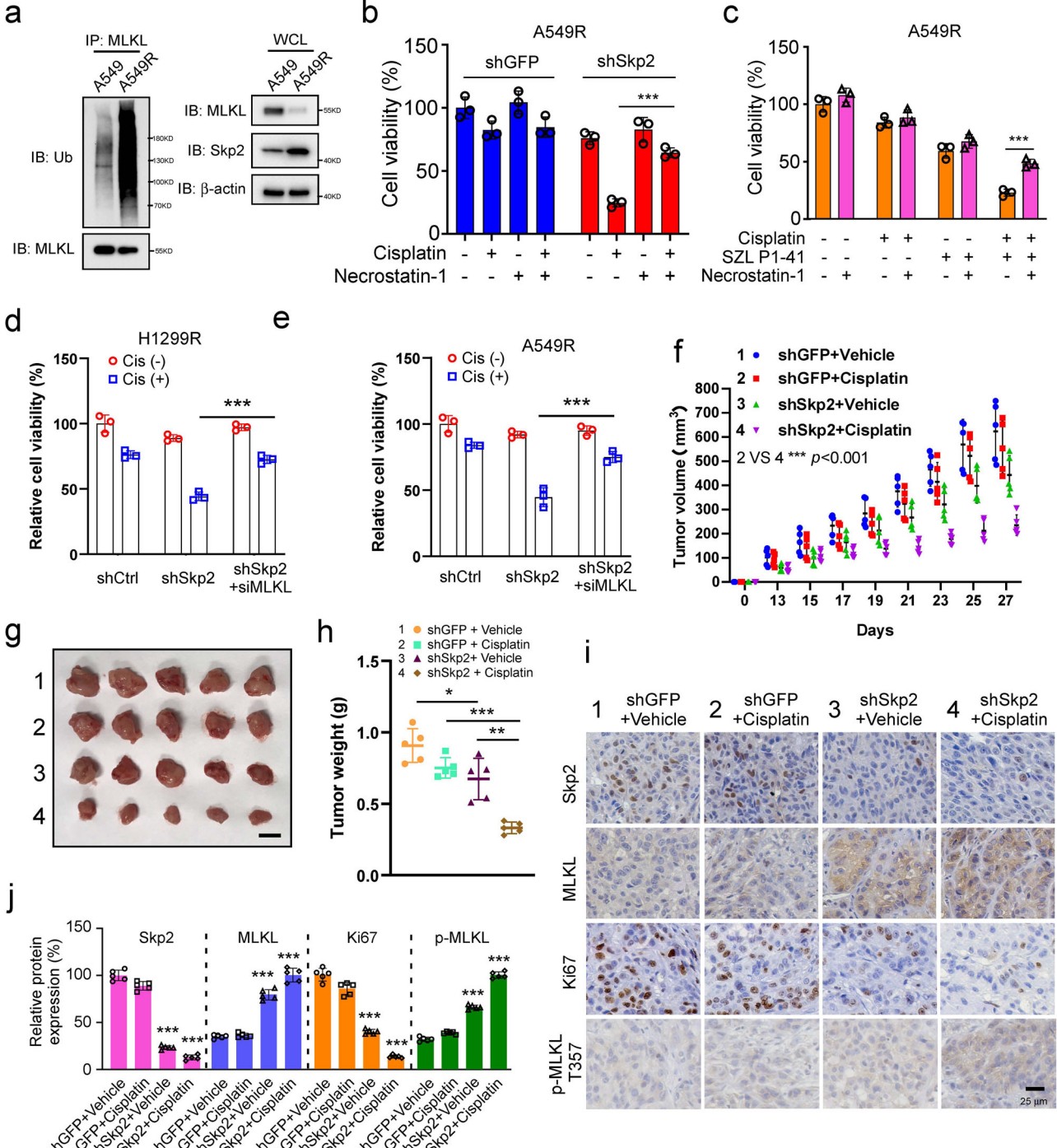

**Fig. 7 Skp2-ubiquitinated MLKL degradation confers cisplatin-resistant of NSCLC cells in vivo. a** The parental A549 cells and cisplatin-resistant A549R cells were treated with MG132 for 6 h, WCEs were prepared and subjected to MLKL ubiquitination analysis. **b** Cisplatin-resistant A549R stable cells were pretreated with 50 μM Necrostatin-1 for 1 h and then treated with DMSO or cisplatin (20 μM) for 48 h. Cell viability was measured. Data represent mean ± SD from three independent experiments. ***$p < 0.001$ by Student's $t$ test. **c** Cisplatin-resistant A549R cells were pretreated with 50 μM Necrostatin-1 for 1 h and then treated with DMSO, cisplatin (20 μM), SZL P1-41 (10 μM) or a cisplatin and SZL P1-41combination for 48 h. Cell viability was measured. Data represent mean ± SD from three independent experiments. ***, $p < 0.001$ by Student's $t$ test. **d, e** H1299R-shSkp2 and A549R-shSkp2 stable cells were transiently knockdown MLKL by siRNA following without/with cisplatin treatment and conducted cell viability assays. Data represent mean ± SD from three independent experiments. **$p < 0.01$, ***$p < 0.001$ by Student's $t$ test, significant difference compared with the cisplatin-treated Skp2-knockdown cells. **f–h** Tumorigenesis of cisplatin-resistant A549R stable cells treated with vehicle or cisplatin. Tumor volume was recorded (**f**), tumor size was monitored (**g**), and tumors were weighed (**h**). Scale bar, 1 cm. $N = 5$ mice per group. *$p < 0.05$, **$p < 0.01$, ***$p < 0.001$ 1-way ANOVA test with Dunnett's multiple comparisons test. **i, j** IHC staining of Skp2, MLKL, Ki67 and phospho-MLKL Thr357 in xenograft tumors from the vehicle or cisplatin-treated shGFP-A549R and shSkp2-A549R stable cells. Data represent mean ± SD from three independent experiments. Scale bar, 25 μm. ***$p < 0.001$ 1-way ANOVA test with Dunnett's multiple comparisons test.

reason for chemotherapeutic resistance, a major problem in current cancer treatment[61]. Like apoptosis, tumor cells can develop resistance against necroptosis to ensure survival[62]. Our results showed that pre-treatment with necroptosis inhibitor Necrostatin-1 significantly compromised cisplatin-reduced cell viability in Skp2-knockdown (Fig. 7b) and SZL P1-41-treated (Fig. 7c) A549R cells, suggesting that the necroptosis mechanism could be involved in NSCLC cells' resistance to cisplatin. Therefore, new therapies that induce programmed cell deaths, including apoptosis and necroptosis, still need to be explored. As an effector of necroptosis, MLKL is transcriptionally up-regulated by IFNs in cancer cells, leading to a higher level of MLKL that sensitizes cells towards necroptosis[63]. Genetic or pharmacological inhibition of MLKL significantly rescues PC cells from BV6-, BV6/TNFα-, or 2'3'-cGAMP/BV6/zVAD.fmk-induced cell death[64,65]. MLKL upregulation by ID1 overexpression sensitizes NSCLC cells to gefitinib treatment via inducing necroptotic cell death[66]. Inhibition of MLKL activity by necrosulfonamide (NSA) significantly attenuated cisplatin-induced cell death in cisplatin resistance HepG2/DDP cells overexpressing RIP3, indicating that MLKL contributed to cisplatin-triggered HepG2/DDP-RIP3 cells death[67]. These studies suggest that altering the MLKL protein level or activity might be valuable for sensitizing cells towards necroptotic cell death and overcoming cancer cell apoptosis resistance. Our results indicated that Skp2 was increased in cisplatin-resistant H1299R and A549R cells, accompanied by decreased MLKL (Fig. 6b). Similarly, Skp2 is increased in paclitaxel-resistant prostate cancer cells[68] and the proteasome inhibitor bortezomib, carfilzomib or ixazomib-resistant multiple myeloma cells[69]. We also provided evidence that Skp2 regulates MLKL through ubiquitin-mediated proteasomal degradation (Fig. 5, Supplementary Fig. 1), suggesting that, at least in part, Skp2 upregulation is responsible for the downregulation of MLKL in NSCLC cells with acquired resistance to cisplatin. Our results showed that the upregulation of Skp2 was also accompanied by the decreased RIP1 and RIP3 in cisplatin-resistance H1299R and A549R cells (Fig. 6b). Recent research reported that somatic epigenetic silencing of RIP3 inactivates necroptosis and contributes to chemoresistance in malignant mesothelioma[70]. Therefore, we could not exclude the possibility that RIP1 and/or RIP3 are regulated by Skp2 and responsible for cisplatin resistance in NSCLC cells, and the mechanisms underlying how Skp2 regulates RIP1 and RIP3 need to be further explored.

Recent studies indicate that MLKL undergoes multiple forms of degradative and non-degradative ubiquitin modifications with different effects on necroptosis[71–74]. MLKL is K63-ubiquitinated at K51, K77, K172 and K219, requiring RIPK3-induced phosphorylation during necroptosis and ubiquitylation simultaneously. K219 ubiquitination protects mice from necroptosis-induced skin damage and renders cells resistant to MLKL-mediated cell death[73]. Human MLKL K230, which corresponds with murine MLKL residue K219, has been reported as a ubiquitination site[75]. Additionally, proteasome inhibitors reportedly facilitate the stabilization of MLKL, supporting the potential role of proteasome-mediated degradation of MLKL[72]. Interestingly, necroptosis-specific multi-mono-ubiquitylation of MLKL occurs following its activation and oligomerization[71]. Our results showed that Skp2 regulated MLKL stability and promoted MLKL ubiquitination and degradation via proteasome (Fig. 5, Supplementary Fig. 1), suggesting that Skp2 is involved in modifying MLKL with K48-ubiquitination. Although the molecular mechanisms and cellular consequences of MLKL ubiquitination are starting to emerge, many fundamental questions, such as the E3 ligases

responsible for MLKL mono- or poly-ubiquitination, need elucidation.

The present study links Skp2 with a key component of necroptosis complex MLKL to cisplatin resistance in NSCLC. The study provides a proof of concept to target Skp2 as a strategy to overcome therapeutic resistance and provides the rationale for developing novel drug combinations to enhance the efficacy of cisplatin-based therapies in NSCLC. In summary, we demonstrate that the Skp2-MLKL axis might contribute to cisplatin resistance through inhibiting cancer cell death in NSCLC. Our study offers convincing evidence that the pharmacological inactivation of Skp2 with current chemotherapeutic agents is a promising approach for treating NSCLC patients with failed prior treatments.

## Methods

**Ethical considerations.** This study was approved by the Ethics Committee of the Second Xiangya Hospital or the Ethics Committee of Xiangya Hospital, Central South University, China. All the patients provided their written informed consent. The animal experiments were approved by the Medical Research Animal Ethics Committee, Central South University, China. Ethical approval and informed consent were obtained to fulfill the institutional requirements.

**Reagents and antibodies.** Cisplatin and chemical reagents, including Tris, NaCl, SDS and DMSO, for molecular biology and buffer preparation were purchased from Sigma-Aldrich (St. Louis, MO, USA). Cell culture media and supplements were from Invitrogen (Grand Island, NY). Skp2 inhibitor SZL P1-41 (#HY-100237) was purchased from MedChem Express (Monmouth Junction, NJ). Necrostatin-1 (#S8037), Cycloheximide (CHX, #S7418) and MG132 (#S2619) were purchased from Selleck (Houston, TX). Control siRNA (#6568), Skp2 siRNA I (#7753) and Skp2 siRNA II (#7756) were purchased from Cell Signaling Technology (Beverly, MA). MLKL siRNA (sc-93430) was purchased from Santa Cruz (Dallas, TX). Antibodies against Skp2 (#2652, 1:1000), MLKL (#14993, 1:1000), RIP1 (#3493, 1:1000), phospho-RIPK1 Ser166 (#44590, 1:1000), RIP3 (#13526, 1:1000), Ubiquitin (#3936, 1:1000) and p27 (#3686, 1:1000) were obtained from Cell Signaling Technology (Beverly, MA). Antibodies against β-actin (A5316, 1:10000), Flag tag (F3165, 1:1000) and Flag-HRP (A8592, 1:10000) were from Sigma-Aldrich (St. Louis, MO). Antibody against GFP tag (TA150032, 1:1000) was purchased from OriGene (Rockville, MD). MLKL antibody (#GTX107538, 1:1000) was from GeneTex (Irvine, CA). MLKL antibody (#orb32399, 1:1000) was from Biorbyt (St. Louis, MO). RIP3 (NBP1-77299, 1:1000) was from Novus Biologicals (Littleton, CO). Phospho-MLKL Thr357 (#MAB9187, 1:1000) was from R&D Systems Inc. (Minneapolis, MN). Secondary antibodies, including anti-rabbit IgG HRP (#7074, 1:10000) and anti-mouse IgG HRP (#7076, 1:10000) were purchased from Cell Signaling Technology (Beverly, MA). Antibody conjugates were visualized by chemiluminescence (ECL; cat#34076, Thermo Fisher, Waltham, MA).

**Clinical tissue sample collection.** NSCLC tumor tissues and the corresponding adjacent non-tumor tissues for Western blot analysis were obtained from the Department of Thoracic Surgery at the Second Xiangya Hospital with written informed consent ($n = 22$). NSCLC tumor tissues and the corresponding non-tumor adjacent tissues for immunohistochemical staining (IHC) analysis were obtained from the Department of Pathology at Xiangya Hospital and the Second Xiangya Hospital ($n = 39$). All specimens were collected with written informed consent following an Institutional Review Board-approved protocol (2022826) by the Medical Ethics Committee of The Second Xiangya Hospital, Central South University. Patients were diagnosed and classified by the Department of Pathology of Xiangya Hospital and the Second Xiangya Hospital following WHO guidelines. All the patients received no treatment before surgery.

**Cell lines and cell culture.** Human NSCLC cells, including NCI-H23 (CRL-5800™), NCI-H125 (CRL-5801™), NCI-H460 (HTB-177™), NCI-H520 (HTB-182™), NCI-H1299 (CRL-5803™), NCI-H358 (CRL-5807™), NCI-H1650 (CRL-5883™), HCC827 (CRL-2868™) from American Type Culture Collection (ATCC) were cultured in RPMI-1640 medium supplemented with 10% FBS and 1% antibiotics according to the ATCC protocols. Human A549 NSCLC cells, human immortalized bronchial epithelial (HBE) cells, human immortalized lung fibroblasts MRC5 and 293T cells were cultured as previously described[76,77]. Cells were cytogenetically tested and authenticated before being frozen. Each vial of frozen cells was thawed and maintained for 2 months (10 passages). To establish cisplatin-resistant cell lines, H1299 and A549 cells were induced by continuous exposure and gradually increasing concentrations of cisplatin (initially with 1 μM to maximally with 20 μM). The resistant cells were continuously maintained in medium containing 20 μM cisplatin for over 6 months and utilized for subsequent experiments.

**Plasmid constructs, lentiviral infection and transient transfection**. Lentivirus plasmids containing *pLKO.1-shSkp2* were purchased from Thermo Scientific. The *pLKO.1-shGFP* (plasmid #30323), the lentiviral packaging plasmid *psPAX2* (plasmid #12260) and the envelope plasmid *pMD2.G* (plasmid #12259) were available on Addgene (Cambridge, MA). The expression constructs *pCMV6-Entry-Skp2* (#RC214001) and *pCMV6-Entry-MLKL* (#RC213152) were purchased from Ori-Gene (Rockville, MD). The MLKL expression construct *pEGFP-MLKL* was purchased from Youbio (Changsha, CN). The generation of gene stable knocking down NCI-H1299, NCI-H23, NCI-H125 or A549 cell lines was performed as described previously[77]. For transient expression of MLKL or Skp2 by plasmid transfection, cells in 6-well plates were transiently transfected with 2.0 µg construct with Lipofectamine 2000 (#11668-019, Invitrogen, Carlsbad, CA) for 48 h following the manufacturer's instructions. RNA interference was performed as described previously[77]. Cells were grown in 6-well plates and transfected with 100 pmol Control siRNA (#6568), Skp2 siRNA I (#7753), Skp2 siRNA II (#7756) or siMLKL (sc-93430) using HiPerFect transfection reagent (#301705, Qiagen) for 72 h according to the manufacturer's instructions and the knockdown of Skp2 was confirmed by Western blot analysis.

**Protein preparation and Western blotting**. Frozen tissue samples were sectioned into small pieces and dissolved in lysis buffer containing 50 mM Tris-Cl (pH 8.0), 150 mM NaCl, 0.1% SDS, 100 µg/mL phenylmethylsulfonyl fluoride, 2 µg/mL aprotinin, 2 µg/mL leupeptin and 1% NP-40. The samples were homogenized, sonicated and kept on ice for 30 min. After centrifugation, the supernatant was collected for immunoblotting analysis. Cultured cells were harvested, and the whole-cell lysates were prepared. Protein concentration was determined using the BCA Assay Reagent (cat#23228, Pierce, Rockford, IL).

**Real-time quantitative polymerase chain reaction (RT-qPCR)**. RT-qPCR was performed as previously described[78]. The primers for *MLKL* are as follows: Forward sequence: ttcacccataagccaaggag; Reverse sequence: cccagaggacgattccaaag. The primers for *GAPDH* are as follows: Forward sequence: tgttgccatcaatgacccctt; Reverse sequence: ctccacgacgtactcagcg.

**Cell viability assay**. Cells ($2 \times 10^3$ per well) were seeded in 96-well plates and treated with different doses of compounds or DMSO control. Cell viability was examined at various time points using the WST-1 reagent (Roche, Mannheim, Germany) as described previously[77] .

**Anchorage-independent cell growth assay**. Cells ($8 \times 10^3$ per well) were seeded into 6-well plates with 0.3% Basal Medium Eagle agar containing 10% FBS and cultured. The cultures were maintained at 37 °C in a 5% $CO_2$ incubator for 2 or 3 weeks. The colonies were counted under a microscope as previously described[77] .

**Co-immunoprecipitation (Co-IP) assay**. Co-IP assays were performed as described previously[76]. Antibodies were used for immunoprecipitation: MLKL (#GTX107538A, GeneTex), Skp2 (#2652, Cell Signaling Technology) or normal Rabbit IgG (NI01, Calbiochem). Immunocomplexes were resolved by SDS-PAGE and co-immunoprecipitated proteins were detected using Skp2 (#2652, Cell Signaling Technology), and MLKL (#66675-1-Ig, Proteintech), respectively. The VeriBlot for IP Detection Reagent (HRP) (#ab131366, Abcam) was used to avoid interference from denatured IgG heavy and light chains.

**Immunohistochemical staining (IHC)**. The NSCLC tissues and the paired adjacent tissues were fixed, embedded, and subjected to IHC analysis[79] and the immunoreactions were evaluated independently by two pathologists as described previously[23].

**Ubiquitination assay**. For endogenous ubiquitination detection, cells were harvested and lysed with modified RIPA buffer (20 mM NAP, pH7.4, 150 mM NaCl, 1% Triton, 0.5% Sodium-deoxycholate, and 1% SDS) supplemented with protease inhibitors and 10 mM N-Ethylmaleimide (NEM). After sonication, the lysates were boiled at 95 °C for 15 min, diluted with RIPA buffer containing 0.1% SDS, then centrifuged at 4 °C (16000 × g for 15 min). The supernatant was incubated with specific antibody and protein A-Sepharose beads overnight at 4 °C. After extensive washing, bound proteins were eluted with 2 × SDS sample loading buffer and separated on an SDS-PAGE, followed by Western blotting. For the Nickel pull-down assay, cells were pelleted and lysed with lysis buffer (6 M guanidine-HCl, 0.1 M $Na_2HPO_4/NaH_2PO_4$, 0.01 M Tris/HCl, pH 8.0, 5 mM imidazole, and 10 mM β-mercaptoethanol) supplemented with protease inhibitors and 10 mM N-Ethylmaleimide (NEM). After sonication and centrifugation, the supernatant was incubated with 50 mL Ni-NTA-agarose (QIAGEN Inc, Valencia, CA) at room temperature for 4 h. The beads were washed with the following buffers: (1) 6 M guanidine-HCl, 0.1 M Na₂HPO4/NaH₂PO₄, 0.01 M Tris/HCl, pH 8.0, 5 mM imidazole plus 10 mM β-mercaptoethanol; (2) 8 M Urea, 0.1 M $Na_2HPO_4/NaH_2PO_4$, 0.01 M Tris/HCl, pH 8.0, 10 mM imidazole, 10 mM β-mercaptoethanol plus 0.1% Triton X-100; (3) 8 M urea, 0.1 M $Na_2HPO_4/NaH_2PO_4$, 0.01 M Tris/HCl, pH 6.3, 10 mM β-mercaptoethanol (buffer A), 20 mM imidazole plus 0.2% Triton X-100,

(4) buffer A with 10 mM imidazole plus 0.1% Triton X-100; (5) buffer A with 10 mM imidazole plus 0.05% Triton X-100. After the last wash, the proteins were eluted with a 2 × SDS sample loading buffer containing 200 mM imidazole and separated on an SDS-PAGE, followed by Western blotting.

**In vivo tumor growth assay**. All mice were maintained and manipulated according to strict guidelines established by the Medical Research Animal Ethics Committee (20220585), Central South University, China. Cells ($1 \times 10^6$) were s.c. injected into 6-week-old athymic nude mice ($n = 5$) at the right flank to generate the xenograft mouse model. Tumors were measured by caliper every 3 days. Tumor volume was recorded and calculated according to the following formula: tumor volume (mm³) = (length × width × width/2). Mice were monitored until the endpoint. At that time, mice were euthanized and tumors extracted.

**Statistics and reproducibility**. Statistical analyses were performed using the statistical package SPSS (version 16.0 for Windows, SPSS Inc, Chicago, IL, USA) and GraphPad Prism (GraphPad 7.0, San Diego, CA, USA). All quantitative data are expressed as mean values ± S.D of three independent experiments. Differences between means were evaluated by Student's $t$ test or analysis of variance (ANOVA) when data were normally distributed. The $\chi^2$ test was used to evaluate associations between various clinicopathological parameters Skp2 and MLKL. Pearson rank correlation was used for correlation tests. A probability value of $p < 0.05$ was used as the criterion for statistical significance.

**Reporting summary**. Further information on research design is available in the Nature Portfolio Reporting Summary linked to this article.

## Data availability

All data supporting the findings of this study are available from the corresponding author upon reasonable request. Uncropped and unedited blot/gel images are available in Supplementary Fig. 4. Source data are available in Supplementary Data 1.

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

## Acknowledgements

The authors thank John Angles (University at Albany, State University of New York) for editorial assistance. This work was supported by the National Natural Science Foundation of China (Grant No. 82073260, 81972837, 82003203, 81572280, 81401548) and the Natural Science Foundation of Hunan Province (Grant No. 2021JJ31011, 2021JJ41058, 2019JJ40420).

## Author contributions

Conception and design: H.D.L., W.L., X.F.Y., H.L.Z., L.Z.; Development of methodology: H.D.L., W.L., X.F.Y., H.L.Z., L.Z., Q.G., X.Y.H., C.W., L.J.L., J.W.; Acquisition of data: H.D.L., W.L., X.F.Y., H.L.Z., L.Z., Q.G., X.Y.H., C.W., L.J.L., J.W.; Analysis and interpretation of data: H.D.L., W.L., X.F.Y.; Writing, review, and/or revision of the manuscript: H.D.L., W.L., X.F.Y.; Administrative, technical, or material support: H.D.L., W.L., X.F.Y., H.L.Z., L.Z., L.J.L., J.W.; Study supervision: H.D.L., W.L., X.F.Y. All authors read and approved the final manuscript.

## Competing interests

The authors declare no competing interests.
