## [Peer Review File · Communications Biology]

Reviewers' comments:

Reviewer #1 (Remarks to the Author):

I. Brief summary of the manuscript

The present study links Skp2 and a key component of necroptosis complex MLKL to cisplatin resistance in non-small cell lung cancer (NSCLC). The study targeted Skp2 to overcome therapeutic resistance and encourages development of novel drug combinations to enhance the efficacy of cisplatin-based therapies in NSCLC.

II. Overall impression of the work

While the study resonates with the general approach biologists/biochemists often take in such work, there is also an immediate sense of overgeneralization of the problem caused by cisplatin resistance in lung cancer. Although the results of the study appear convincing more detailed description of the study topic and further analysis of the data are needed to make the work more interesting to researchers, clinicians and drug developers alike.

III. Specific comments, with recommendations for addressing each comment

1. The introduction is far too short and not even remotely precise, so needs re-writing.

Need to be addressed in the introduction: The authors need to take into consideration that lung cancer has several subtypes. Small cell and non-small cell lung cancer (NSCLC) where NSCLC is divided further, adeno- (LUAD) and squamous cell lung cancers (LUSC) being the most prominent. If it was not enough, LUAD has several driver mutations providing a great variety of targets for more specific therapy. It is also a mistake that cisplatin is the main therapy in NSCLC of any type. In the latest therapy guideline, immunotherapy has great significance even in the advanced stages of the disease. Not to mention, cisplatin is rarely used in monotherapy, and recently even in stages III and IV carboplatin is the one that is more frequently used in therapy due to the systemic toxicity of cisplatin that many patients cannot tolerate.

2. While someone reading the manuscript finds the topic novel, it was enough to search the internet for "SKP2 in cancer" to find a published paper (see below):

Ting Wu, Xinsheng Gu, and Hongmei Cui: Emerging Roles of SKP2 in Cancer Drug Resistance. *Cells*. 2021 May; 10(5): 1147. Published online 2021 May 10. doi: 10.3390/cells10051147

Having skimmed the manuscript, it has become clear, that inhibition of SKP2 can block drug resistance of a great variety of cancer medications and in almost any cancer and any cancer pathways. It appears that cisplatin is just an additional drug that was not included in the manuscript above.

3. Nevertheless, the conclusions for cisplatin can still be novel, if structured better and reanalyzed taking lung cancer subtypes and driver mutations into consideration. All the data need to be presented with that specificity in mind. Including the cancer cell lines used in the study (e.g. A549 is a dominantly KRAS mutant cell line). Patient NSCLC subtypes and mutation analysis are also important.

4. Methodology, statistics and presentation of data appear fine.

5. General inhibitors of single molecules have been attempted in clinical applications before to overcome drug resistance. Some have even reached clinical trials. As an example, I'd like to highlight inhibitors of drug transporters, that are generally present in our cells. Systemic use of such drugs, however, has led to the death of patients due to general toxicity. Hence, I am always wary to see single molecules in cancer therapy that once inhibited are hailed to eliminate all types of cancer cells. I would suggest toning down the conclusions.

Reviewer #2 (Remarks to the Author):

The authors investigate the contribution of control of mixed-lineage kinase domain-like (MLKL) protein levels to sensitivity of non-small cell lung cancers to platinum-based chemotherapy. The manuscript employs an array of approaches that includes human samples and in vitro and in vivo techniques. Questions and points for the authors to consider are listed below to help improve the manuscript:

- Introduction: please note, chemotherapy is not the primary or only treatment for lung cancer and is also employed in an adjuvant early-stage setting. Please modify and broaden the first paragraph to acknowledge the role of chemotherapy in the lung cancer management landscape.
- The authors rightly assess Skp2 and MLKL expression in a panel of cell lines. Please provide a rationale for why the four cell lines were chosen for further in vitro evaluation (Fig. 1e-g) and why H23 cells were chosen for in vivo evaluation (Fig. 1h-j).
- Given the inverse observation of survival outcomes for high MLKL and high Skp2, it would be worthwhile to confirm whether, in publicly available RNAseq, an inverse correlation in Skp2 and MLKL levels also exists.
- Fig. 3d: changes in MLKL level following Skp2 inhibition are not completely clear. Can an alternative repeat be included and densitometry performed?
- An Skp2-MLKL association between both ectopic and endogenous protein was observed by immunoprecipitation (Fig. 4). It would be worthwhile showing this is the case in other cell lines. Do you also expect the Skp2 inhibitor SZL P1-41 to block the association between these proteins? Furthermore, given the overall focus on platinum-based chemotherapy, it would benefit the study's findings to perform these IPs also in the presence/absence of cisplatin and in the resistance setting (perhaps to be shown in Fig. 6).
- Please perform MG132 degradation experiments in the presence of ectopic Skp2 expression (rather than depletion – Fig. 5a). This would more adequately demonstrate an impact on MLKL protein degradation.
- Fig. 5c-d: unfortunately it isn't completely clear to me there's any marked change in Ub-MLKL following respective overexpression of Skp2 or inhibition with SZL P1-41. Could you include another repeat or perform densitometry on these blots? The change in ubiquitinated MLKL is clearer in Fig. 5e.
- Fig. 6: given the title and focus of the manuscript is upon MLKL degradation conferring resistance to platinum agents, does ectopic expression of MLKL, or at least, it's specific blockade of degradation, enhance cisplatin sensitivity in platinum resistant cell lines? Is there any change in Skp2 or MLKL levels following an acute exposure to platinum agents across cell lines? Are MLKL levels altered following siRNA-mediated Skp2 depletion in the resistant cell lines (Fig. 6c)? Lines 162-168 indicate that the Skp2 inhibitor specifically upregulate MLKL levels. While SZL P1-41 is suggested to upregulate MLKL protein levels, blockade of Skp2 would also impact degradation of its other substrates. Please perform more direct experiments (e.g. ectopic MLKL in resistance cells, or MLKL depletion in parental cells) to suggest a role for MLKL in platinum resistance. Might MLKL depletion in Skp2 depleted cells enhance cisplatin sensitivity?
- Fig. 7: could the authors provide evidence that necrostatin-1 is functioning appropriately (e.g. impacted RIPK1 phosphorylation)? Can the authors investigate the in vivo induction of necroptosis following Skp2 depletion and cisplatin treatment (e.g. via IHC methods)?
- Overall: The link between Skp2 and MLKL levels have been investigated. However, suggesting that this association, as indicated in the manuscript's title, confers resistance to platinum-based chemotherapy remains less clear. I suggest modifying the manuscript title more accurately reflect the study findings. Moreover, line 311 points to the Skp2-MLKL as a major determinant of cisplatin resistance. Please rewrite this to more accurately reflect the study findings suggest that this axis might contribute to resistance rather than be a major player.
- Figures and figure legends: please include how many times each experiment was performed and statistics details within all figure legends. Please also include error bars where appropriate (e.g. Fig. 5b, 6a).

Reviewer #3 (Remarks to the Author):

The authors describe in their manuscript the identification and characterization of a new interaction of Skp2, an E3-ubiquitin ligase and MLKL a necroptosis regulator and the functional consequences of the genetic or pharmacologic modulation of Skp2 in NSCLC cells in vitro and in vivo. Additionally they analyzed clinical samples for the expression of both genes. They could prove a contribution of Skp2 overexpression to (i) tumorigenic properties of NSCLC cells and (ii) the downregulation of MLKL protein level by Skp2-mediated ubiquitinylation. Moreover, they obtained data pointing to a possible contribution to Cisplatin resistance. The manuscript is well written and most of the presented data are of high quality. The manuscript is of interest for both basic and translational research scientists as it describes a new mechanism for the regulation of necroptosis potentially with clinical relevance.

However, the reviewer sees some limitations that should be addressed by the authors:

Major remarks

1. The most important point of this work, from the translational/clinical view, is the proposed influence of Cisplatin resistance by Skp2 overexpression. However, the real influence of the clinical treatment is not analyzed in the manuscript. The authors tried to analyze available gene expression data for Skp2 and MLKL for the prognostic value using the online tool KM-plotter (Fig. 2e). The combination of several genome wide expression data sets enables the analysis of >1000 patients and revealed a prognostic value of Skp2 and MLKL expression. However, if the single data sets were analyzed by the reviewer, most do not support this result. Thus, the stated prognostic value is questionable and may originate from different patient cohorts with largely different survival/gene expression results. The authors must improve this in-silico analysis to exclude any bias. Additionally, it would support the influence of Cisplatin resistance by the Skp2-MLKL pathway if the authors could present data from public gene expression datasets comparing platinum sensitive and resistant patients.
2. The data for the Skp2-mediated ubiquitinylation and degradation of MLKL are not completely convincing (Fig. 5). Specifically Fig. 5c, d do not show a convincing difference of Ub-MLKL levels. How many replicates were done and what were the results? The number of replicates should be stated for all experiments in the figure legends.
3. The Skp2 knockdown influenced the cell viability of parental and resistant cells under Cisplatin treatment similarly (albeit significant for resistant cells only). The used Cisplatin concentration (20 μ M) had a different effect on parental and resistant cells (\geq IC50 vs. $<$ IC50). It would be interesting if parental cells, treated with a similarly less effective dosis (ca. 2 μ M) would respond similar to the Skp2 knockdown. Additionally, the authors show a reduced viability of Cisplatin resistant cells after Skp2 inhibition and an increased effect for the combination with Cisplatin (Fig. 6 f,g). Is this an additive or synergistic effect? What is the effect for the parental cells? This would also be interesting for the Necrostatin-1 treatment (Fig. 7b, c). To certainly prove an active role of Skp2 in causing Cisplatin resistance overexpression studies should be done.
4. The literature references should be updated to include several missing publications related to cisplatin resistance and Skp2 or MLKL function. These data should also be discussed. Additionally, the authors should correct some general statements that are proven for certain tumor entities only and carefully check the cited literature in the method section.
5. The English language should be checked by a native speaker.
6. The manuscript should be checked to include correct statements supported by the shown data instead of indirect conclusions (e.g. lines 95/6; 166/7).
7. The authors should discuss a limitation of their study – albeit they prove an interaction of Skp2 and MLKL they do not prove that this interaction is important for the observed effects. Other targets of Skp2 may contribute to these results.

Minor remarks

1. The authors state, that the resistant cell cultures were cultured in medium containing 20 μ M

Cisplatin. Was this exposure stopped before the described experiments were done or were the resistant cells under continuous exposure? If stopped, what was the time span between medium switch and the experiments?

2. In the material and method section the authors state about the analysis of clinical samples from n=22 patients (line 343) and n=39 patients (line 403). Were these different patients? Please clarify.

3. The authors show contrary protein expression changes between adjacent and tumor tissue for Skp2 (up in T) and MLKL (down in T; Fig. 1d). Some sample pairs do not show a changed expression between the tissues. It would be of interest if these pairs for Skp2 and MLKL are from the same or different patients.

4. Is the labeling of Fig. 3a correct and represent the two right lanes cells without Myc-Skp2? Or is the opposite correct? What is the protein expression in native 293T?

5. The authors mention the cell line MRC5 but did not include it in the method section. Please add the data.

April 26, 2023

Dr. Georgios Giamas

Communications Biology

RE: MS ID#: COMMSBIO-22-3273

Title: Skp2-ubiquitinated MLKL degradation confers cisplatin-resistant in non-small cell lung cancer cells

Dear Editors-in-Chief,

We would like to thank the Reviewers for taking the time and effort to review the manuscript. We appreciate the critiques and suggestions and have revised the manuscript. Here is a point-by-point response to the reviewers' comments and concerns.

We hope that you will agree that we have addressed the editor's and reviewer's concerns point by point and that this manuscript can now be considered for publication in *Communications Biology*.

Sincerely,

Haidan Liu, Dr. P.H.

Department of Cardiovascular Surgery

Clinical Center for Gene Diagnosis and Therapy

The Second Xiangya Hospital

Central South University

139 Renmin Road

Changsha 410011, Hunan, China

Reviewers' comments:

Reviewer #1 (Remarks to the Author):

I. Brief summary of the manuscript

The present study links Skp2 and a key component of necroptosis complex MLKL to cisplatin resistance in non-small cell lung cancer (NSCLC). The study targeted Skp2

to overcome therapeutic resistance and encourages development of novel drug combinations to enhance the efficacy of cisplatin-based therapies in NSCLC.

II. Overall impression of the work

While the study resonates with the general approach biologists/biochemists often take in such work, there is also an immediate sense of overgeneralization of the problem caused by cisplatin resistance in lung cancer. Although the results of the study appear convincing more detailed description of the study topic and further analysis of the data are needed to make the work more interesting to researchers, clinicians and drug developers alike.

III. Specific comments, with recommendations for addressing each comment

1. The introduction is far too short and not even remotely precise, so needs re-writing.

Need to be addressed in the introduction: The authors need to take into consideration that lung cancer has several subtypes. Small cell and non-small cell lung cancer (NSCLC) where NSCLC is divided further, adeno- (LUAD) and squamous cell lung cancers (LUSC) being the most prominent. If it was not enough, LUAD has several driver mutations providing a great variety of targets for more specific therapy. It is also a mistake that cisplatin is the main therapy in NSCLC of any type. In the latest therapy guideline, immunotherapy has great significance even in the advanced stages of the disease. Not to mention, cisplatin is rarely used in monotherapy, and recently even in stages III and IV carboplatin is the one that is more frequently used in therapy due to the systemic toxicity of cisplatin that many patients cannot tolerate.

Response: *We have revised the Introduction Section. The changes have been highlighted in the revised version.*

2. While someone reading the manuscript finds the topic novel, it was enough to search the internet for “SKP2 in cancer” to find a published paper (see below):

Ting Wu, Xincheng Gu, and Hongmei Cui: Emerging Roles of SKP2 in Cancer Drug Resistance. *Cells*. 2021 May; 10(5): 1147. Published online 2021 May 10. doi: 10.3390/cells10051147

Having skimmed the manuscript, it has become clear, that inhibition of SKP2 can block drug resistance of a great variety of cancer medications and in almost any cancer and any cancer pathways. It appears that cisplatin is just an additional drug that was not included in the manuscript above.

Response: *Thanks for your comments. Skp2 has been reportedly involved in a great variety of cancer types' drug resistance, including cisplatin resistance, which is also mentioned in the above paper published by Ting Wu et al. We have reported that depletion of Skp2 sensitized*

nasopharyngeal carcinoma (NPC) cells to cisplatin treatment, highlighting a promising approach for Skp2 targeting therapy in NPC treatment¹. Our present study tries to elucidate the different mechanisms of cisplatin resistance in NSCLC. We discovered that Skp2 ubiquitinates MLKL to inhibit necroptosis and that MLKL is a novel Skp2 target protein.

3. Nevertheless, the conclusions for cisplatin can still be novel, if structured better and reanalyzed taking lung cancer subtypes and driver mutations into consideration. All the data need to be presented with that specificity in mind. Including the cancer cell lines used in the study (e.g. A549 is a dominantly KRAS mutant cell line). Patient NSCLC subtypes and mutation analysis are also important.

Response: *We have revised the Introduction Section. The changes have been highlighted in the revised version.*

4. Methodology, statistics and presentation of data appear fine.

Response: *Thanks for your comment.*

5. General inhibitors of single molecules have been attempted in clinical applications before to overcome drug resistance. Some have even reached clinical trials. As an example, I'd like to highlight inhibitors of drug transporters, that are generally present in our cells. Systemic use of such drugs, however, has led to the death of patients due to general toxicity. Hence, I am always wary to see single molecules in cancer therapy that once inhibited are hailed to eliminate all types of cancer cells. I would suggest toning down the conclusions.

Response: *Thanks for your comment. We have toned down the conclusion as "Skp2-regulated MLKL ubiquitination and degradation, at least partially, contributes to cisplatin-resistant in NSCLC cells" in the revised version.*

Reviewer #2 (Remarks to the Author):

The authors investigate the contribution of control of mixed-lineage kinase domain-like (MLKL) protein levels to sensitivity of non-small cell lung cancers to platinum-based chemotherapy. The manuscript employs an array of approaches that includes human samples and in vitro and in vivo techniques. Questions and points for the authors to consider are listed below to help improve the manuscript:

- Introduction: please note, chemotherapy is not the primary or only treatment for lung cancer and is also employed in an adjuvant early-stage setting. Please modify and broaden the first paragraph to acknowledge the role of chemotherapy in the lung cancer management landscape.

Response: *We have revised the Introduction Section. The changes have been highlighted in the revised version.*

- The authors rightly assess Skp2 and MLKL expression in a panel of cell lines. Please provide a rationale for why the four cell lines were chosen for further in vitro evaluation (Fig. 1e-g) and why H23 cells were chosen for in vivo evaluation (Fig. 1h-j).

Response: *When we constructed cisplatin-resistant cell lines, A549 and H1299 were most likely to develop acquired resistance, so we chose A549/A549R and H1299/H1299R cells for further study. H358 and H460 cells were selected as representatives of Skp2 low-expression cells. The H23 cell was used as a representative cell for Skp2 high expression. We therefore generated Skp2 knockdown stable cells and performed the in vitro and in vivo studies.*

- Given the inverse observation of survival outcomes for high MLKL and high Skp2, it would be worthwhile to confirm whether, in publicly available RNAseq, an inverse correlation in Skp2 and MLKL levels also exists.

Response: *We analyzed the relationship between MLKL and SKP2 RNA expression in TCGA-LUAD. The result showed that the Pearson correlation coefficient of MLKL and SKP2 was 0.244 ($p < 0.05$), which indicated that there might be a slightly positive relationship between MLKL and SKP2 in patients with LUAD. SKP2 ubiquitination of MLKL is a post-translational modification at the protein level, so there will be correlation discrepancies between the protein level and the RNA level. Moreover, no public database is available to evaluate the relationship in relapse samples with cisplatin treatment. It is not contradicted by the survival analysis results because the survival analysis was performed based on patients grouped by expression levels of MLKL or SKP2. The correlation results may differ if comparing high- or low-expressed groups. Unfortunately, we cannot prove this hypothesis because cut-off values for grouping are currently unavailable.*

- Fig. 3d: changes in MLKL level following Skp2 inhibition are not completely clear. Can an alternative repeat be included and densitometry performed?

Response: *We have provided the new data of Fig. 3d and performed the densitometry analysis in the revised version (Fig. 3e, f).*

- An Skp2-MLKL association between both ectopic and endogenous protein was observed by immunoprecipitation (Fig. 4). It would be worthwhile showing this is the case in other cell lines. Do you also expect the Skp2 inhibitor SZL P1-41 to block the association between these proteins? Furthermore, given the overall focus on platinum-based chemotherapy, it would benefit the study's findings to perform these IPs also in the presence/absence of cisplatin and in the resistance setting (perhaps to be shown in Fig. 6).

Response: *We performed co-immunoprecipitation (Co-IP) to examine the association between Skp2 and MLKL in A549 cells as well as the effect of Skp2 inhibitor SZL P1-41 on the Skp2-MLKL association. The result indicated that endogenous MLKL and Skp2 interaction was found in A549 cells. The Skp2 inhibitor SZL P1-41 could not substantially block the Skp2-MLKL association in A549 cells. The new data have been added to the revised manuscript (See Fig. 4e,4f and 4g in the revised version). We also assessed endogenous MLKL and Skp2 interaction in the presence/absence of cisplatin via Co-IP in both A549R and A549 cells. The endogenous association between MLKL and Skp2 was diminished in both A549R and A549 cells treated with cisplatin. The new data have been included in the revised version (See Fig. 6f, 6g in the revised version).*

- Please perform MG132 degradation experiments in the presence of ectopic Skp2 expression (rather than depletion – Fig. 5a). This would more adequately demonstrate an impact on MLKL protein degradation.

Response: *We ectopically expressed Skp2 and conducted MG132 degradation experiment in A549 cells. The result demonstrated that ectopic Skp2 expression decreased endogenous MLKL protein level, which was restored by treatment with the proteasome inhibitor MG132 in A549 cells. The new data have been included in the revised version (See Fig. 5b in the revised version).*

- Fig. 5c-d: unfortunately it isn't completely clear to me there's any marked change in Ub-MLKL following respective overexpression of Skp2 or inhibition with SZL P1-41. Could you include another repeat or perform densitometry on these blots? The change in ubiquitinated MLKL is clearer in Fig. 5e.

Response: *We repeated the experiment and the new data have been added to the revised manuscript (See Fig. 5d in the revised version).*

- Fig. 6: given the title and focus of the manuscript is upon MLKL degradation conferring resistance to platinum agents, does ectopic expression of MLKL, or at least, it's specific blockade of degradation, enhance cisplatin sensitivity in platinum

resistant cell lines? Is there any change in Skp2 or MLKL levels following an acute exposure to platinum agents across cell lines? Are MLKL levels altered following siRNA-mediated Skp2 depletion in the resistant cell lines (Fig. 6c)? Lines 162-168 indicate that the Skp2 inhibitor specifically upregulate MLKL levels. While SZL P1-41 is suggested to upregulate MLKL protein levels, blockade of Skp2 would also impact degradation of its other substrates. Please perform more direct experiments (e.g. ectopic MLKL in resistance cells, or MLKL depletion in parental cells) to suggest a role for MLKL in platinum resistance. Might MLKL depletion in Skp2 depleted cells enhance cisplatin sensitivity?

Response: *We ectopically expressed MLKL following without/with cisplatin treatment and conducted cell viability assays in H1299R and A549R cells. The results demonstrated that ectopic MLKL expression enhanced cisplatin-reduced cell viability in both H1299R and A549R cells, suggesting that MLKL is involved in cisplatin resistance in NSCLC. The new data have been included in the revised version (See Fig. 6h, 6i in the revised version). Upon acutely exposure to cisplatin, the Skp2 protein level decreased was accompanied by the MLKL protein level slightly increased in parental A549 cells, whereas Skp2 and MLKL protein levels did not obviously change in A549R cells. The new data have been included in the revised version (See Fig. S2 in the revised version). The WB results indicated that siRNA-mediated Skp2 depletion was accompanied by MLKL protein level increases in both H1299R and A549R cells. The new data have been included in the revised version (See Fig. 6c in the revised version). In order to confirm the role of MLKL in cisplatin resistance further, we transiently knockdown MLKL by siRNA following without/with cisplatin treatment and performed cell viability assays in H1299R-shSkp2 and A549R-shSkp2 stable cells. The results indicated that the knockdown of MLKL compromised cisplatin-reduced cell viability in both H1299R-shSkp2 and A549R-shSkp2 stable cells, confirming the critical role of MLKL in cisplatin resistance in NSCLC. The new data have been included in the revised version (See Fig. 7d, 7e in the revised version).*

- Fig. 7: could the authors provide evidence that necrostatin-1 is functioning appropriately (e.g. impacted RIPK1 phosphorylation)? Can the authors investigate the in vivo induction of necroptosis following Skp2 depletion and cisplatin treatment (e.g. via IHC methods)?

Response: *It has been reported that the phosphorylation of several sites, including Ser14, Ser15, Ser161 or Ser166 is critical for RIPK1 kinase activity and sensitive to necrostatin-1 treatment². We thus checked the phospho-Ser166 level upon necrostatin-1 treatment. The WB result showed that the RIPK1 inhibitor necrostatin-1 significantly abolished the phosphorylation of RIPK1 at Ser166 in A549 cells, indicating that necrostatin-1 is functioning appropriately. The new data have been included in the revised version (See Fig. S3 in the revised version). The phosphorylation of MLKL at Thr357 is critical for necroptosis³. We performed IHC assay to investigate the in vivo induction of necroptosis following Skp2 depletion and cisplatin treatment. The IHC result revealed that xenograft tumors from cisplatin-treated Skp2-knockdown A549R cells exhibited a significant increase in the phosphorylation level of MLKL at Thr357. The new data have been included in the revised version (See Fig. 7i in the revised version).*

- Overall: The link between Skp2 and MLKL levels have been investigated. However, suggesting that this association, as indicated in the manuscript's title, confers resistance to platinum-based chemotherapy remains less clear. I suggest modifying the manuscript title more accurately reflect the study findings. Moreover, line 311 points to the Skp2-MLKL as a major determinant of cisplatin resistance. Please rewrite this to more accurately reflect the study findings suggest that this axis might contribute to resistance rather than be a major player.

Response: *Thank you for your suggestions. We modified the manuscript title as "Skp2-ubiquitinated MLKL degradation confers cisplatin-resistant through inhibiting necroptosis in non-small cell lung cancer cells". We rewrote the statement in line 311 as "we demonstrate that the Skp2-MLKL axis might contribute to cisplatin resistance through inhibiting cancer cell death in NSCLC".*

- Figures and figure legends: please include how many times each experiment was performed and statistics details within all figure legends. Please also include error bars where appropriate (e.g. Fig. 5b, 6a).

Response: *We have included the experiment details within the figure legends and included error bars in Fig. 5b, Fig. 6a and Fig. S1b.*

Reviewer #3 (Remarks to the Author):

The authors describe in their manuscript the identification and characterization of a new interaction of Skp2, an E3-ubiquitin ligase and MLKL a necroptosis regulator and the functional consequences of the genetic or pharmacologic modulation of Skp2 in NSCLC cells in vitro and in vivo. Additionally they analyzed clinical samples for the expression of both genes. They could prove a contribution of Skp2 overexpression to (i) tumorigenic properties of NSCLC cells and (ii) the downregulation of MLKL protein level by Skp2-mediated ubiquitinylation. Moreover, they obtained data pointing to a possible contribution to Cisplatin resistance. The manuscript is well written and most of the presented data are of high quality. The manuscript is of interest for both basic and translational research scientists as it describes a new mechanism for the regulation of necroptosis potentially with clinical relevance.

However, the reviewer sees some limitations that should be addressed by the authors:

Major remarks

1. The most important point of this work, from the translational/clinical view, is the proposed influence of Cisplatin resistance by Skp2 overexpression. However, the real

influence of the clinical treatment is not analyzed in the manuscript. The authors tried to analyze available gene expression data for Skp2 and MLKL for the prognostic value using the online tool KM-plotter (Fig. 2e). The combination of several genome wide expression data sets enables the analysis of >1000 patients and revealed a prognostic value of Skp2 and MLKL expression. However, if the single data sets were analyzed by the reviewer, most do not support this result. Thus, the stated prognostic value is questionable and may originate from different patient cohorts with largely different survival/gene expression results. The authors must improve this in-silico analysis to exclude any bias. Additionally, it would support the influence of Cisplatin resistance by the Skp2-MLKL pathway if the authors could present data from public gene expression datasets comparing platinum sensitive and resistant patients.

Response: We found no public RNA-seq datasets of patients with NSCLC providing clinical information on cisplatin (CDDP)-resistance. A gene expression profiling, with 133 frozen JBR.10 tumor samples, including 62 observation [OBS] and 71 adjuvant CDDP/vinorelbine (ACT)⁴, was downloaded from the Gene Expression Omnibus (GEO) repository. Moreover, the expression level of SKP2 (lack of MLKL expression data) was compared in the OBS group with the ACT group and groups based on survival status, including alive, lung cancer-caused death and other condition-caused death. The results showed no significant differences between these groups (Figure R1A, B), while the median level of SKP2 expression was relatively low in the alive group, indicating that NSCLC patients may benefit from low SKP2 expression. It is consistent with the hypothesis of prognostic value in our manuscript, although we could not validate it directly due to the lack of public database of NSCLC patients with CDDP-resistance. Furthermore, we analyzed the expression of SKP2 and MLKL in datasets of CDDP-resistant NSCLC animal (GSE135720) and cell models (GSE108139⁵ and GSE48244) to find more evidence to support our results. In GSE135720, the SKP2 expression is down-regulated in the CDDP-resistance group, with the MLKL expression up-regulated (without significance, Figure R1C, D), which contradicts our results. However, this dataset does not provide the gene expression count profiling, and Principal Component Analysis (PCA) results show no significant difference between samples in different groups (Figure R1E). It is remarkable that the results of different gene expression analyses (Resistant vs. Sensitive) both in GSE108139 (H23 and H460 cells) (Figure R1F) and GSE48244 (U1810 cells) (Figure R1G) in accord with our results that the SKP2 expression is up-regulated and the MLKL expression is down-regulated in CDDP-resistant cells.

Figure R1. (A-D) Beeswarm plots of SKP2 and MLKL expression levels in different groups. (E) The PCA plot of samples in CDDP-resistance mouse group and control group. (F, G) The results of differential gene expression analysis in GSE108139 (F) and GSE48244 (G).

2. The data for the Skp2-mediated ubiquitylation and degradation of MLKL are not completely convincing (Fig. 5). Specifically Fig. 5c, d do not show a convincing

difference of Ub-MLKL levels. How many replicates were done and what were the results? The number of replicates should be stated for all experiments in the figure legends.

Response: *We repeated the experiment and the new data have been added to the revised manuscript (See Fig. 5d in the revised version).*

3. The Skp2 knockdown influenced the cell viability of parental and resistant cells under Cisplatin treatment similarly (albeit significant for resistant cells only). The used Cisplatin concentration (20 μ M) had a different effect on parental and resistant cells (\geq IC50 vs. <

Response: *The parental sensitive cells which express a relatively lower protein level of Skp2 are sensitive to cisplatin, and 20 μ M cisplatin could induce cell death. In contrast, in the acquired resistant cells, a high protein level of Skp2 confers chemoresistance and knockdown of Skp2 restores the sensitivity to cisplatin. Our in vitro and in vivo data suggested that Skp2 is required for cisplatin resistance in NSCLC cells.*

4. The literature references should be updated to include several missing publications related to cisplatin resistance and Skp2 or MLKL function. These data should also be discussed. Additionally, the authors should correct some general statements that are proven for certain tumor entities only and carefully check the cited literature in the method section.

Response: *High level of Skp2 confers cisplatin resistance in nasopharyngeal carcinoma cells¹ and mantle cell lymphoma cells⁶. Upregulation of Skp2 reportedly enhanced cisplatin resistance in A549 cells⁷. However, the role of Skp2 in the cisplatin resistance of NSCLC has not been fully elucidated. Inhibition of MLKL activity by necrosulfonamide (NSA) significantly attenuated cisplatin-induced cell death in cisplatin resistance HepG2/DDP cells overexpressing RIP3, indicating that MLKL contributed to cisplatin-triggered HepG2/DDP-RIP3 cells death⁸. We have added these in the revised version. In addition, We have checked and revised the cited literature in the method section. The changes have been highlighted in the revised version.*

5. The English language should be checked by a native speaker.

Response: *We have asked a native English speaker John Angles (University at Albany, State University of New York) to edit this manuscript.*

6. The manuscript should be checked to include correct statements supported by the shown data instead of indirect conclusions (e.g. lines 95/6; 166/7).

Response: *We have re-written and made the statements directly (lines 95/6; 166/7 in the previous version).*

7. The authors should discuss a limitation of their study – albeit they prove an interaction of Skp2 and MLKL they do not prove that this interaction is important for the observed effects. Other targets of Skp2 may contribute to these results.

Response: *The M phase and G2 phase are the most sensitive stages to radiotherapy or DNA damage agents. Knockdown of Skp2 induces p27 expression and cell cycle arrest which may enhance the anti-tumor effect of cisplatin by DNA damage. We have added this to the Discussion Section in the revised version.*

Minor remarks

1. The authors state, that the resistant cell cultures were cultured in medium containing 20 μ M Cisplatin. Was this exposure stopped before the described experiments were done or were the resistant cells under continuous exposure? If stopped, what was the time span between medium switch and the experiments?

Response: *The resistant cells were continuously exposed for three days every two weeks to maintain the resistance. This exposure was stopped for one week before the described experiments.*

2. In the material and method section the authors state about the analysis of clinical samples from n=22 patients (line 343) and n=39 patients (line 403). Were these different patients? Please clarify.

Response: *Sorry for making confuse. The samples are obtained from different patients. We have clarified this in the revised version.*

3. The authors show contrary protein expression changes between adjacent and tumor tissue for Skp2 (up in T) and MLKL (down in T; Fig. 1d). Some sample pairs do not show a changed expression between the tissues. It would be of interest if these pairs for Skp2 and MLKL are from the same or different patients.

Response: *We found that the samples from some patients showed high Skp2 and MLKL expression, and some showed low Skp2 and MLKL expression. We analyzed the protein*

expression of *Skp2* and *MLKL* in tumor tissues and the matched adjacent tissues, and there was a negative correlation between the expression of *SKP2* and *MLKL* in most patients (Fig.2d). Our data showed that *MLKL* is a substrate of *Skp2*, and ubiquitination of *MLKL* by *Skp2* confers chemoresistance of NSCLC cells. The *Skp2-MLKL* axis is a potential target for chemosensitization.

4. Is the labeling of Fig. 3a correct and represent the two right lanes cells without Myc-Skp2? Or is the opposite correct? What is the protein expression in native 293T?

Response: We apologize for the mistakes. We already checked the whole manuscript throughout and have corrected the errors in the revised version.

5. The authors mention the cell line MRC5 but did not include it in the method section. Please add the data.

Response: We apologize for the mistakes. We have included the MRC5 cell line in the method section.

References

1. Yu, X. et al. Skp2-mediated ubiquitination and mitochondrial localization of Akt drive tumor growth and chemoresistance to cisplatin. *Oncogene*. **38**, 7457-7472 (2019).
2. Ofengeim, D.&Yuan, J. Regulation of RIP1 kinase signalling at the crossroads of inflammation and cell death. *Nat Rev Mol Cell Biol*. **14**, 727-736 (2013).
3. Sun, L. et al. Mixed lineage kinase domain-like protein mediates necrosis signaling downstream of RIP3 kinase. *Cell*. **148**, 213-227 (2012).
4. Zhu, C.Q. et al. Prognostic and predictive gene signature for adjuvant chemotherapy in resected non-small-cell lung cancer. *J Clin Oncol*. **28**, 4417-4424 (2010).
5. Vera, O. et al. An epigenomic approach to identifying differential overlapping and cis-acting lncRNAs in cisplatin-resistant cancer cells. *Epigenetics*. **13**, 251-263 (2018).
6. Yan, W., Yang, Y.&Yang, W. Inhibition of SKP2 Activity Impaired ATM-Mediated DNA Repair and Enhanced Sensitivity of Cisplatin-Resistant Mantle Cell Lymphoma Cells. *Cancer Biother Radiopharm*. **34**, 451-458 (2019).
7. Hou, Q. et al. FAM60A promotes cisplatin resistance in lung cancer cells by activating SKP2 expression. *Anticancer Drugs*. **31**, 776-784 (2020).
8. Zhang, B. et al. Receptor interacting protein kinase 3 promotes cisplatin-induced necroptosis in apoptosis-resistant HepG2/DDP cells. *Neoplasma*. **66**, 694-703 (2019).

Reviewers' comments:

Reviewer #2 (Remarks to the Author):

The authors have addressed my concerns. Thank-you for considering and amending my suggested changes.

Reviewer #3 (Remarks to the Author):

The authors describe in their manuscript the identification and characterization of a new interaction of Skp2, an E3-ubiquitin ligase and MLKL a necroptosis regulator and the functional consequences of the genetic or pharmacologic modulation of Skp2 in NSCLC cells in vitro and in vivo. Additionally they analyzed clinical samples for the expression of both genes. They could prove a contribution of Skp2 overexpression to (i) tumorigenic properties of NSCLC cells and (ii) the downregulation of MLKL protein level by Skp2-mediated ubiquitinylation. Moreover, in the revised version they included data pointing to a contribution of the Skp2-MLKL interaction to Cisplatin resistance. The manuscript is well written and most of the presented data that are of high quality. The manuscript is of interest for both basic and translational research scientists as it describes a new mechanism for the regulation of necroptosis potentially with clinical relevance.

Albeit the authors responded well to most remarks, the reviewer still sees limitations including some that were not addressed by the authors in the revised version:

Major remarks

1. The authors tried to analyze available gene expression data for Skp2 and MLKL for the prognostic value using the online tool KM-plotter (Fig. 2e). The combination of several genome wide expression data sets enables the analysis of >1000 patients and revealed a prognostic value of Skp2 and MLKL expression. However, if the single data sets were analyzed by the reviewer, most do not support this result. Thus, the stated prognostic value is questionable and may originate from different patient cohorts with largely different survival/gene expression results. The authors did neither comment on this point nor adapted their analysis, thus they still must improve this in-silico analysis to exclude any bias.
2. In Fig. 6 the authors show data for Skp2 and MLKL protein expression in cisplatin sensitive and resistant cell lines. There are pronounced differences between 6b vs. 6f, g (WCL) for both Skp2 and MLKL level? Whereas 6b shows strong expression differences between sensitive and resistant cells this is not the case for the whole cell lysates 6f vs. g. What is the reason?

Reviewers' comments:

Reviewer #3 (Remarks to the Author):

The authors describe in their manuscript the identification and characterization of a new interaction of Skp2, an E3-ubiquitin ligase and MLKL a necroptosis regulator and the functional consequences of the genetic or pharmacologic modulation of Skp2 in NSCLC cells in vitro and in vivo. Additionally they analyzed clinical samples for the expression of both genes. They could prove a contribution of Skp2 overexpression to (i) tumorigenic properties of NSCLC cells and (ii) the downregulation of MLKL protein level by Skp2-mediated ubiquitylation. Moreover, in the revised version they included data pointing to a contribution of the Skp2-MLKL interaction to Cisplatin resistance. The manuscript is well written and most of the presented data that are of high quality. The manuscript is of interest for both basic and translational research scientists as it describes a new mechanism for the regulation of necroptosis potentially with clinical relevance.

Albeit the authors responded well to most remarks, the reviewer still sees limitations including some that were not addressed by the authors in the revised version:

Major remarks

1. The authors tried to analyze available gene expression data for Skp2 and MLKL for the prognostic value using the online tool KM-plotter (Fig. 2e). The combination of several genome wide expression data sets enables the analysis of >1000 patients and revealed a prognostic value of Skp2 and MLKL expression. However, if the single data sets were analyzed by the reviewer, most do not support this result. Thus, the stated prognostic value is questionable and may originate from different patient cohorts with largely different survival/gene expression results. The authors did neither comment on this point nor adapted their analysis, thus they still must improve this in-silico analysis to exclude any bias.

Response: *The online tool KM-plotter provides a simple and available approach to analyze and visualize the prognostic value of genes in NSCLC by combining gene expression data and survival information, especially from GEO, EGA and TCGA databases. One of the major benefits of integrating several data sets is the increased sample size, which makes the hypothesis test results more accurate. However, it comes with some problems. The integration of data sets from different platforms is not recommended. Thus, we further improved the analysis by combining data sets from the GPL570, including GSE29013 (n=55), GSE37745 (n=196), GSE31210 (n=79), and GSE157011 (n=483). GSE157011 is not included in the KM-plotter, whereas we exclude GSE19188 (included in the KM-plotter, n=156) because the expression data has been normalized, leading to negative numbers. Then Kaplan–Meier analysis was performed to show the correlation between gene expression and overall survival of patients with NSCLC. The results show that patients with low Skp2 (n=244) expression or high MLKL (n=410) expression may have better outcomes, consistent with the results from the KM-plotter. This result was updated in Fig. 2e. The changes*

have been highlighted in the revised version. The “limma” R package provides several approaches to remove the batch effects, especially between data sets from the same platform. Furthermore, some researchers recently proposed a novel algorithm for consolidated analysis to correct nonbiological effects across technologies (Ref: Tang K, Ji X, Zhou M, Deng Z, Huang Y, Zheng G, Cao Z. Rank-in: enabling integrative analysis across microarray and RNA-seq for cancer. *Nucleic Acids Res.* 2021 Sep 27;49(17):e99. doi: 10.1093/nar/gkab554. PMID: 34214174; PMCID: PMC8464058.). Although mRNA expression profiling reveals the underlying prognostic value of genes, proteomics is also expected to be applied since post-translational modifications play crucial roles in protein function and stability regulation, and the changes at mRNA levels are not entirely match that in protein levels.

Data of patients with NSCLC (n=813) consisted of GSE29013, GSE37745, GSE31210, and GSE157011. Patients were divided into two groups: low and high SKP2 expression groups or low and high MLKL expression groups. Kaplan-Meier curves were plotted for correlation between two groups and overall survival (OS).

2. In Fig. 6 the authors show data for Skp2 and MLKL protein expression in cisplatin sensitive and resistant cell lines. There are pronounced differences between 6b vs. 6f, g (WCL) for both Skp2 and MLKL level? Whereas 6b shows strong expression differences between sensitive and resistant cells this is not the case for the whole cell lysates 6f vs. g. What is the reason?

Response: We compared the expression of Skp2 in parental cells (A549, H1299) and cisplatin resistant cells (A549R, H1299R), and found that Skp2 was highly expressed in resistant cells, but MLKL was downregulated (Fig. 6b). We also assessed endogenous MLKL and Skp2 interaction in the presence/absence of cisplatin via Co-IP in A549R (Fig. 6f) and A549 cells (Fig. 6g). The WCL (input) in Fig 6f showed the protein level of Skp2 and MLKL in A549R cells with or without cisplatin treatment, and found that acute exposure to cisplatin slightly changed MLKL expression.

However, Skp2 was unaffected. A similar result was also observed in A549 cells (Fig. 6g).